# Sequential roles for red blood cell binding proteins enable phased commitment to invasion for malaria parasites

Melissa N. Hart ®[1,2], Franziska Mohring[1], Sophia M. DonVito ®[1], James A. Thomas ®[1], Nicole Muller-Sienerth ®[3], Gavin J. Wright[3,4], Ellen Knuepfer ®[2,5], Helen R. Saibil ®[6] & Robert W. Moon ®[1]✉

Invasion of red blood cells (RBCs) by *Plasmodium* merozoites is critical to their continued survival within the host. Two major protein families, the Duffy binding-like proteins (DBPs/EBAs) and the reticulocyte binding like proteins (RBLs/RHs) have been studied extensively in *P. falciparum* and are hypothesized to have overlapping, but critical roles just prior to host cell entry. The zoonotic malaria parasite, *P. knowlesi*, has larger invasive merozoites and contains a smaller, less redundant, DBP and RBL repertoire than *P. falciparum*. One DBP (DBPα) and one RBL, normocyte binding protein Xa (NBPXa) are essential for invasion of human RBCs. Taking advantage of the unique biological features of *P. knowlesi* and iterative CRISPR-Cas9 genome editing, we determine the precise order of key invasion milestones and demonstrate distinct roles for each family. These distinct roles support a mechanism for phased commitment to invasion and can be targeted synergistically with invasion inhibitory antibodies.

Malaria threatens almost half the globe, with six species of *Plasmodium* parasites causing significant disease in humans[1]. Symptoms result from the blood stage of the parasite's life cycle when motile merozoites invade red blood cells (RBCs), multiply within them, and then burst out (egress) to release new invasive merozoites. Since blocking invasion prevents parasite replication, a detailed understanding of how merozoites invade RBCs is essential to develop clinical interventions, such as vaccines and antibody therapies.

Video microscopy studies of *P. knowlesi (Pk)*, and later *P. falciparum (Pf)*, revealed merozoite invasion is a rapid process (~60 s) with several morphologically-defined steps[2,3]. First, the merozoite and RBC form a weak and reversible binding interaction, which quickly progresses into a stronger interaction causing the RBC to deform and 'wrap' around the merozoite. This may facilitate 're-orientation' with realigning its apical end that houses specialised secretory organelles, micronemes and rhoptries, in perpendicular apposition to the RBC

surface. Work in *Pf* identified a further step: fusion between the merozoite's apex and the RBC before internalisation, visualised as a fluorescent signal at the parasite-host cell interface when RBCs are preloaded with a calcium-sensitive fluorescent dye[4]. Then, the merozoite actomyosin motor actively drives the parasite into the parasitophorous vacuole formed by invagination of the RBC membrane[5]. Invasion usually results in RBC echinocytosis, triggered by secretion of rhoptry contents or by perturbation of RBC homeostasis[4]. Within minutes the RBC returns to its biconcave shape, marking the end of a successful invasion[2,3]

The micronemes and rhoptries facilitate each step of invasion by sequentially releasing proteins that act as ligands for distinct receptors. Some of these ligands have been well characterised in *Pf*. For example, PfAMA-1 binds to PfRON2, its parasite-derived receptor inserted into the RBC plasma membrane, to form the moving junction, a molecular seal between merozoite and host cell that 'travels' backwards over the

[1]Department of Infection Biology, Faculty of Infectious and Tropical Disease, London School of Hygiene and Tropical Medicine, London WC1E 7HT, UK. [2]Department of Pathobiology and Population Sciences, Royal Veterinary College, Hawkshead Lane, Hatfield AL9 7TA, UK. [3]Wellcome Sanger Institute, Hinxton, Cambridge CB10 1SA, UK. [4]Department of Biology, Hull York Medical School, York Biomedical Research Institute, University of York, Wentworth Way, York YO10 5DD, UK. [5]Malaria Parasitology Laboratory, Francis Crick Institute, London NW1 1AT, UK. [6]ISMB, Biological Sciences, Birkbeck, University of London, Malet St, London WC1E 7HX, UK. ✉e-mail: Rob.Moon@lshtm.ac.uk

merozoite during internalisation[6,7]. Upon junction formation, the merozoite is presumed to be irreversibly committed to invasion.

Two key families of ligands predicted to mediate steps preceding junction formation are the reticulocyte binding-like (RBL/Rh) proteins and the Duffy binding proteins (DBP/EBA)[4,8]. *Pf* can express up to four DBP and five RBL orthologs. Simultaneously inhibiting PfRBL/DBP-receptor interactions prevents moving junction formation, RBC deformation and potentially merozoite re-orientation[4,9,10]. However, significant redundancy within and possibly between the two families has hampered the dissection of their precise roles. Of all the PfRBL/DBPs, only PfRh5, an atypical RBL ligand restricted to *Plasmodium* species belonging to the subgenus Laverania appears to be essential for invasion, and likely performs a role distinct from the other RBL/DBPs at the point of merozoite-RBC fusion[4,11]. It remains unclear whether the other *Pf* RBL and DBP family members perform the same or distinct functions[12–14].

In contrast, evidence suggests that the RBL and DBP ligands of non-Laveranian Plasmodium *spp*, including the zoonotic parasite, *Pk* and closely related *P. vivax* (Pv), may have distinct functions. *Pk* expresses three DBP ligands (PkDBPα, PkDBPβ, and PkDBPγ), all highly similar in sequence to the single *Pv* DBP ligand, PvDBP. PkDBPβ and PkDBPγ are not required for invasion of human RBCs. However, PkDBPα and PvDBP are essential for invasion of human RBCs and bind to the Duffy antigen receptor for chemokines (DARC), limiting both species to replication in Duffy positive RBCs[15–18]. For *Pv*, RBL ligands interact with reticulocyte-specific receptors, such as PvRBP2b binding to TfR1 (Transferrin receptor 1), limiting this species to invasion of reticulocytes[19]. *Pk* is not restricted to reticulocytes and can propagate in normocytes with one of its two RBL ligands, normocyte binding protein Xa (NBPXa) essential for growth in human RBCs[20]. Thus, for both Pk and Pv, at least one RBL and one DBP are required for invasion of human RBCs. Early studies in *Pk* demonstrated that merozoites could deform Duffy negative cells, but not invade them, and that DBPα null merozoites cannot form a moving junction. However, it is unclear to what extent DBPα null merozoites can deform host cells, and which other steps of invasion are affected[15,21]. Even less is known about the role of Pk/Pv RBL ligands, except that while NBPXa-null parasites can bind to the outside of human RBCs, they fail to invade them[20]

Recently we showed that merozoites can glide across RBC surfaces and that this active movement likely underpins RBC deformation[22]. However, the mechanisms by which gliding merozoites deform RBCs and transition into subsequent steps of invasion are unknown. Here we take advantage of the large size and clear polarity of *Pk* merozoites to explore the early steps of invasion and identify key milestones in host cell commitment. This provides a framework to investigate the roles of the PkRBL and PkDBP families using live microscopic analysis of fluorescently-tagged and conditional knockout (cKO) DBPα and NBPXa parasite lines, revealing these proteins have distinct locations once released from the micronemes. Finally, we demonstrate that NBPXa is essential for host cell deformation, while DBPα has a distinct function downstream of this, but ahead of rhoptry secretion and reorientation. These distinct roles provide a mechanism for staged commitment to host cell selection and invasion and identify these proteins as synergistic targets for therapeutic intervention.

## Results

### Deformation is associated with gliding motility and invasion success

We first used live-cell imaging of *Pk* merozoite-RBC interactions to delineate the morphological stages of invasion. From 20 egresses, 49% merozoite-RBC interactions ($N = 134/275$) displayed gliding motility across the RBC surface (Fig. 1a). Gliding was accompanied by RBC deformation in 67% of these interactions. On average this began within 2 s of gliding onset (median = 1 s, 91 interactions) (Fig. 1a and Supplementary Fig. 1A) and was characterized by a pinching or 'wrapping' of

the host cell membrane across the width of the merozoite (Fig. 1b). This pinching appeared to originate at the merozoite apex and progressed towards the posterior (Supplementary movie 1 from 2.1–9.0 s). There were 5 unclear events (Fig. 1a), but we observed no clear instance of deformation without gliding, corroborating previous work showing gliding motility is required for deformation[22]. Importantly, all invasions ($N = 40$) were preceded by deformation, and as previously shown for *Pf* invasion success[4] was positively correlated with deformation strength (Fig. 1b). These data suggest that both gliding and deformation are critical pre-cursors to *Pk* invasion.

While deformation strength is predictive of invasion success, it is unclear what factors determine a merozoite's ability to initiate deformation. *Pk* merozoites frequently contact many human RBCs before commitment to invasion[22]. The deformation scores of interactions occurring prior to interactions resulting in invasion ('intermediate interactions' in Fig. 1b) are comparable to those for interactions made by merozoites which never invade ('Non-invader events'). Fewer 'intermediate interactions' are observed when *Pk* merozoites invade macaque RBCs, which are highly permissive to *Pk* invasion[22]—suggesting that deformation strength is an indicator of host cell 'suitability' (for example, receptor availability or deformability), rather than merozoite 'readiness' to invade. A significant fraction of gliding interactions on human RBCs (44/134 events) did not progress to deformation (e.g., blue arrow in Supplementary movie 2; Fig. 1a). Thus, whilst gliding is required for deformation, deformation of RBC membranes is not a prerequisite for traversal of RBC surfaces.

### Merozoite apex-RBC fusion precedes re-orientation

We examined individual invasion events to determine precisely when *Pk* merozoites are irreversibly attached to RBCs or 'committed' to invasion. *Pf* merozoites have been described to "pause" for several seconds on the host cell surface as deformation subsides but before internalisation begins[3]. During this stage, a merozoite-RBC fusion event takes place, visualised as a fluorescent signal appearing at the interface between the merozoite apex and Fluo-4-AM loaded RBCs[4]. However, the much smaller size and the more spherical morphology of *Pf* merozoites makes it difficult to determine when this step occurs relative to reorientation and junction formation.

Analysis of events immediately prior to internalisation showed that *Pk* merozoites appear to cease gliding motility—and thus forward movement across the host cell—as deformation subsides and a median 4 s ($N = 30$ events; Supplementary Fig. 1B) prior to the RBC returning to its biconcave morphology. At this point, the stationary merozoite lies lengthwise across the RBC surface, with its apical end firmly pinned to the host cell surface (Fig. 1c, panel 1; Supplementary movie 1 at -9 s). The posterior end of the merozoite then dissociates from the RBC surface, while the merozoite pivots from the apical end until perpendicular to the RBC membrane (Supplementary movie 1 from 9.5–11.5 s and Fig. 1c, panels 2–4). Reorientation consistently began immediately after deformation ceased, and the RBC had returned to its original morphology ($N = 28$ events). Therefore deformation does not mechanically facilitate re-orientation as previously proposed[23,24], but rather deformation and re-orientation are distinct, sequential steps.

Next, we examined merozoites invading Fluo-4-AM loaded RBCs. Strikingly, for all invasions where reorientation could be clearly visualised ($N = 17/18$ events from 12 egress events), a punctate fluorescent signal appeared at the apex of each merozoite 'pinned' to the RBC, a median 2 s before onset of reorientation (Fig. 1d; Supplementary movie 3 at 14 s; Supplementary Fig. 1C). Frequently, we observed this punctate fluorescent signal move into the RBC with the invading parasite (e.g. Supplementary movie 3 from 46 s onwards), suggesting that this signal arises primarily from Fluo-4-AM travelling into calcium-rich merozoite apical compartments[4]. For 5/18 events, a faint but detectable fluorescent signal spread briefly to the RBC cytosol; however, this phenomenon was far less apparent than has been reported

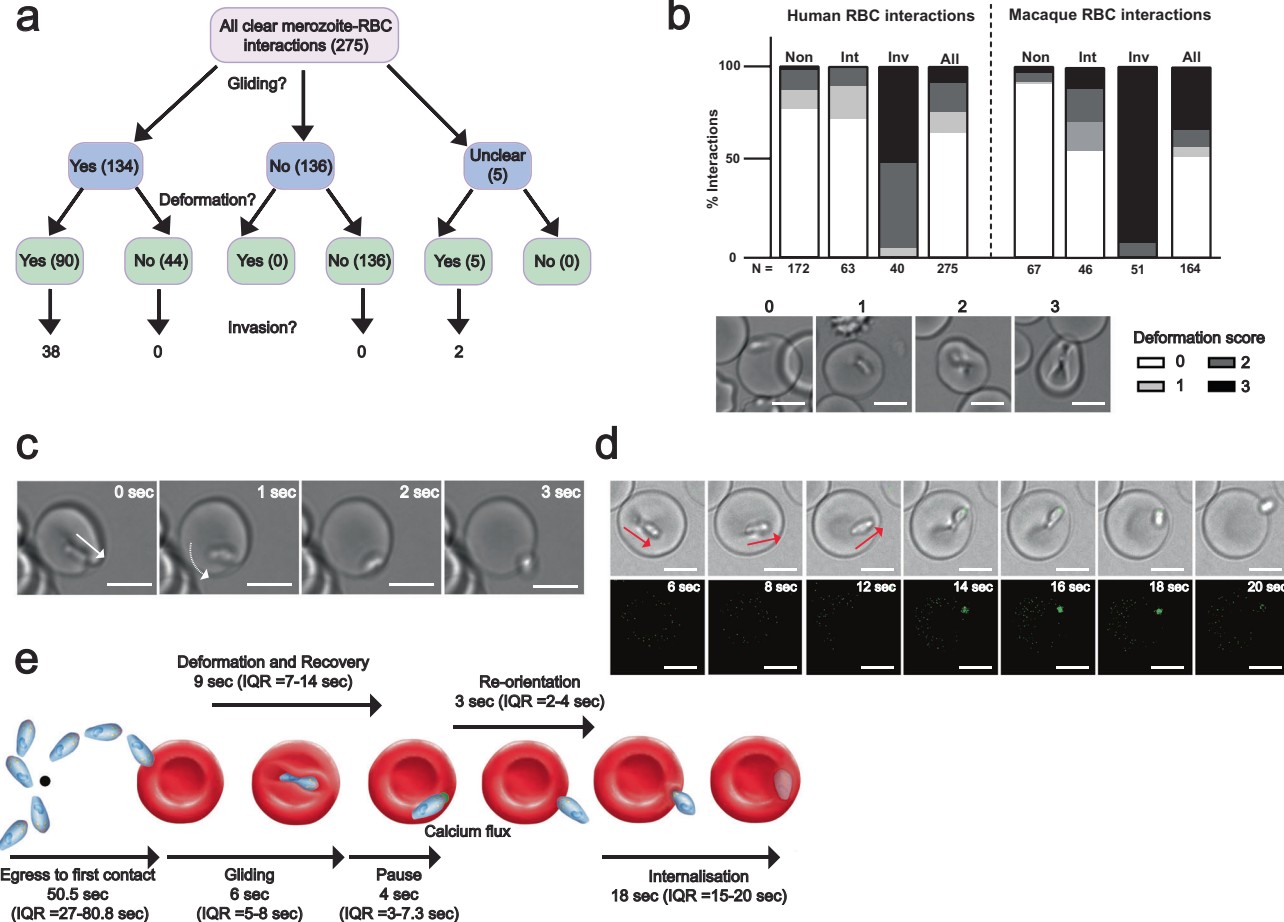

**Fig. 1 | _P. knowlesi_ as a model to delineate steps of RBC invasion. a** Flow chart shows outcomes of all merozoite-human RBC interactions from 20 schizonts, 0–200 s post egress. 'Interaction' = RBC contact lasting ≥2 s. Gliding interaction = forward movement across RBC surface ≥5 s. Events per category indicated in parentheses. **b** RBC deformation scores for human (left) or macaque (right) interactions based on extent of merozoite indentation/wrapping. 0 = no deformation; 1 = shallow indentation/membrane pinching; 2 = deeper indentation to the side of RBC/intermediate level of host cell membrane pinching around parasite; 3 = full host cell wrapping around merozoite. Bar chart indicates a breakdown of deformation scores for interactions from merozoites which never invade (non), intermediate interactions from merozoites that will invade on subsequent contacts

('Int'), interactions leading directly to invasion (Inv), and all merozoite-RBC interaction (all). **c** Supplementary movie 1 stills. Panel 1 shows RBC recovering from deformation and merozoite's apical end firmly attached to the RBC membrane. White arrow tip indicates merozoite apex. Panels 2-4 show the merozoite pivoting on apical end (white curved arrow), until re-orientation is complete. **d** Supplementary movie 3 stills. _Pk_ merozoite displays a fluorescent signal between apex and Fluo-4-AM loaded RBC as gliding (red arrows = direction) comes to a pause (panel 4), but before re-orientation begins (panel 6). **e** Summary schematic depicting order and timings of each step of invasion. For all images, scale bars = 5 μm. Source data for (**b**) and (**e**) are provided as a Source Data file.

for _Pf_[4,25]. Notably, we did not observe any detriment to invasion when human RBCs were pre-loaded with the calcium chelator, BAPTA-AM, in keeping with data for _P. falciparum_ invasions[4] (Supplementary Fig. 1G). This result may suggest that a calcium influx into the host cell, whether parasite derived or coming from the surrounding medium, may not be required for invasion, as others have hypothesized[4,26]. However, we observed no merozoite detachment after detection of Fluo-4-AM signal or reorientation. This indicates that commitment to invasion, and potentially also to moving junction formation, occurs before reorientation and thus earlier than suggested by data from _Pf_ (Fig. 1e)[4,25] and furthermore, that Fluo-4 signal can be used as a reliable marker for such commitment.

**Propensity for strong deformation events underpin macaque RBC invasion success**

We next investigated how _Pk_ invasion and host cell selection differ between human and macaque RBCs. Whilst there was no significant difference in the length of time _Pk_ merozoites spent invading either RBC (median = 32 s vs 35 s from first contact to completion of

internalisation; _p_ = 0.976; Supplementary Fig. 1D), merozoites spent slightly longer deforming macaque vs human RBCs (median = 12.5 s vs 9 s; _p_ = 0.024; Supplementary Fig. 1E), and slightly less time actively entering macaque vs human cells (median internalisation = 15 s vs 18 s; _p_ = 0.011; Supplementary Fig. 1F).

Notably, a much higher proportion of interactions with macaque RBCs resulted in the strongest (score 3) deformation (_N_ = 54/164 events for macaque cells versus 22/275 events for human) (Fig. 1b; Supplementary movie 4). For both host cell types, strong deformation proved to be a reliable predictor of invasion success—with 87% of macaque (47/54 events) and 91% (20/22 events) of human RBC score 3 interactions progressing to invasion. This greater propensity for strong deformation suggests a greater proportion of macaque RBCs are amenable to invasion (e.g., RBC receptor availability, membrane tension etc.).

**Deletion of NBPXa prevents growth of _Pk_ parasites in human but not macaque RBCs**

Having established the morphological events leading to host cell entry, we sought to determine the role of NBPXa during invasion. Our

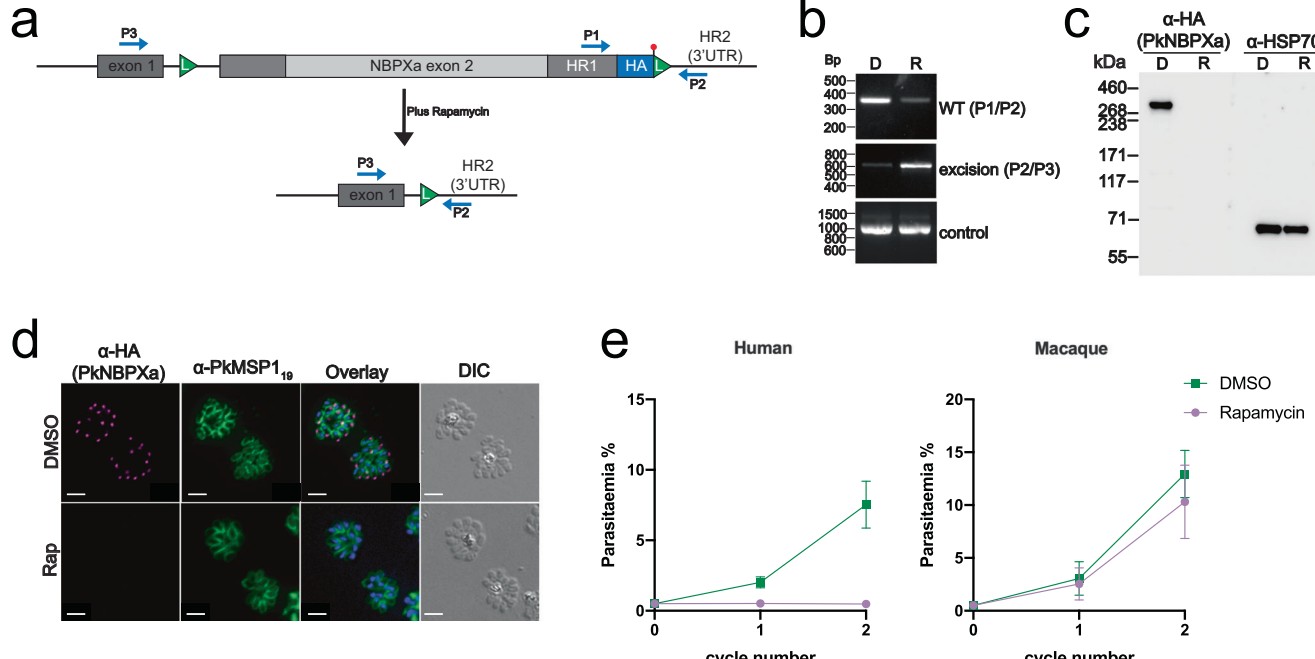

**Fig. 2 | Efficient Rapamycin-induced excision of floxed *NBPXa*. a** Schematic depicting excision of 8.5kbp fragment between LoxP sites (L) upon Rapamycin (Rap) treatment of NBPXa cKO parasites. Diagnostic primer positions noted in blue (P1, P2, P3). **b** Diagnostic PCRs showing outcome of DMSO (D) or Rap (R) treatment of NBPXa cKO parasites. Primers P1/P2 identified non-excised parasites, P2/P3 detected successful excision and control bands amplify an unrelated locus. **c** Western blot showing loss of ~325 kDa HA tagged NBPXa in Rap treated parasites.

Anti-PfHSP70 used as a loading control. **d** IFAs showing ablation of NBPXa expression for Rap treated parasites. Parasites labelled with a rat anti-HA and rabbit anti-MSP1$_{19}$ as a marker for mature, segmented schizonts. Scale bars = 5 μm. **e** Growth of NBPXa cKO parasites in human (left) or macaque (right) RBCs measured at 24 and 48 h after treatment with Rap or DMSO. Data shown are the mean of 5 independent experiments for human and 2 experiments for macaque cells. Error bars +/- SEM. Source data for (**b**) and (**c**) and (**e**) are provided as a Source Data file.

previously NBPXa knockout line can only be maintained in macaque RBCs[20] so we established a conditional knockout (cKO) line that could be routinely maintained in human RBCs. A parasite line containing a dimerisable Cre recombinase (DiCre) was generated using CRISPR-Cas9[18] to insert a DiCre expression cassette into the *Pkp230p* locus (Supplementary Fig. 2A, B). Adding rapamycin (Rap) to this line results in formation of active Cre recombinase, leading to excision of DNA sequence flanked by *loxP* sites. Subsequently, we iteratively modified this line to integrate a *loxP* sequence within the *nbpxa* intron and a HA tag and *loxP* site at the 3′ end of *nbpxa* (Supplementary Fig. 2A, C; Fig. 2a).

Synchronized ring-stage parasites were treated with 10 nM Rap or carrier (0.005% DMSO) for 3 h and samples were taken at the end of the first cycle (~ 27–28 h post-invasion) and analysed by PCR for successful *nbpxa* excision. Deletion of the floxed 8.4 kb sequence removes most of *nbpxa* along with the C-terminal HA tag sequence, leaving only the signal peptide encoding exon 1 (Fig. 2a). PCR analysis demonstrated correct excision of the floxed *nbpxa* sequence in Rap-treated parasites (Fig. 2b). A fainter band was also detected in the control suggesting some formation of active Cre recombinase in absence of Rap. Full-length HA-tagged NBPXa (~325 kDa) was detected by Western blot of mock-treated samples but was absent after Rap treatment (Fig. 2c). Immunofluorescence (IFA) analysis revealed (Fig. 2d) an overall excision rate of 93.5% (n = 22/342 HA positive cells for Rap vs 258/271 DMSO).

Deletion of *nbpxa* led to a severe growth defect of *Pk* in human but not macaque RBCs (Fig. 2e). This is consistent with previous work and explained by the fact that NBPXb, the only *nbpxa* paralogue in *Pk*, binds macaque but not human RBCs, and thus complements the loss of NBPXa only for macaque RBC invasion[20]. Rap treatment of wild-type parasites (Supplementary Fig. 3A) revealed a minimal toxic effect of the drug, but this was not significant.

## NBPXa null merozoites are motile but fail to deform human RBCs

To understand why NBPXa null parasites fail to invade human RBCs, we examined merozoite-RBC interactions using live microscopy. These cKO merozoites were still capable of gliding across RBCs (Fig. 3a; Supplementary movie 5): the median percentage of gliding merozoites per egress was 85% for both mock-treated and Rap-treated parasites (Fig. 3b; p = 0.5281). This result demonstrates that the merozoites were alive, with essential gliding and secretory systems intact, and that NBPXa is not required for the initial parasite-host cell interaction underpinning motility.

On average, NBPXa null merozoites spent longer gliding on host cells (median = 8 s) than mock-treated parasites (median = 6 s; p = 0.0001; Fig. 3c)—a likely outcome as these merozoites could not invade. However, while gliding was unimpeded, they failed to deform host cells (Fig. 3a, d; Supplementary movie 5). For mock-treated parasites, high scoring (score 2 and 3) deformations made up 17.6% of all interactions (N = 85/483 events). In contrast, no score 3 events were recorded for NBPXa-null parasites and score 2 interactions were less than 2% (N = 8/466 interactions) of all interactions. The proportion of score 1 deformations, which included very minor indentations, caused by collision of merozoites gliding into RBCs, was also significantly reduced (14.6% in NBPXa-null parasites versus 22.2% in mock-treated parasites; p = 0.0126). Therefore, NBPXa appears to be critical for strong deformation of human RBCs.

The lack of deformation correlated with a drastically lower invasion efficiency for NBPXa-null parasites. While 58 merozoites from 20 mock-treated schizonts invaded RBCs, only two invasions were observed for merozoites from 20 Rap-treated schizonts. Since the excision efficiency of the cKO line is about 94%, we conclude that these two invasive merozoites probably had intact NBPXa.

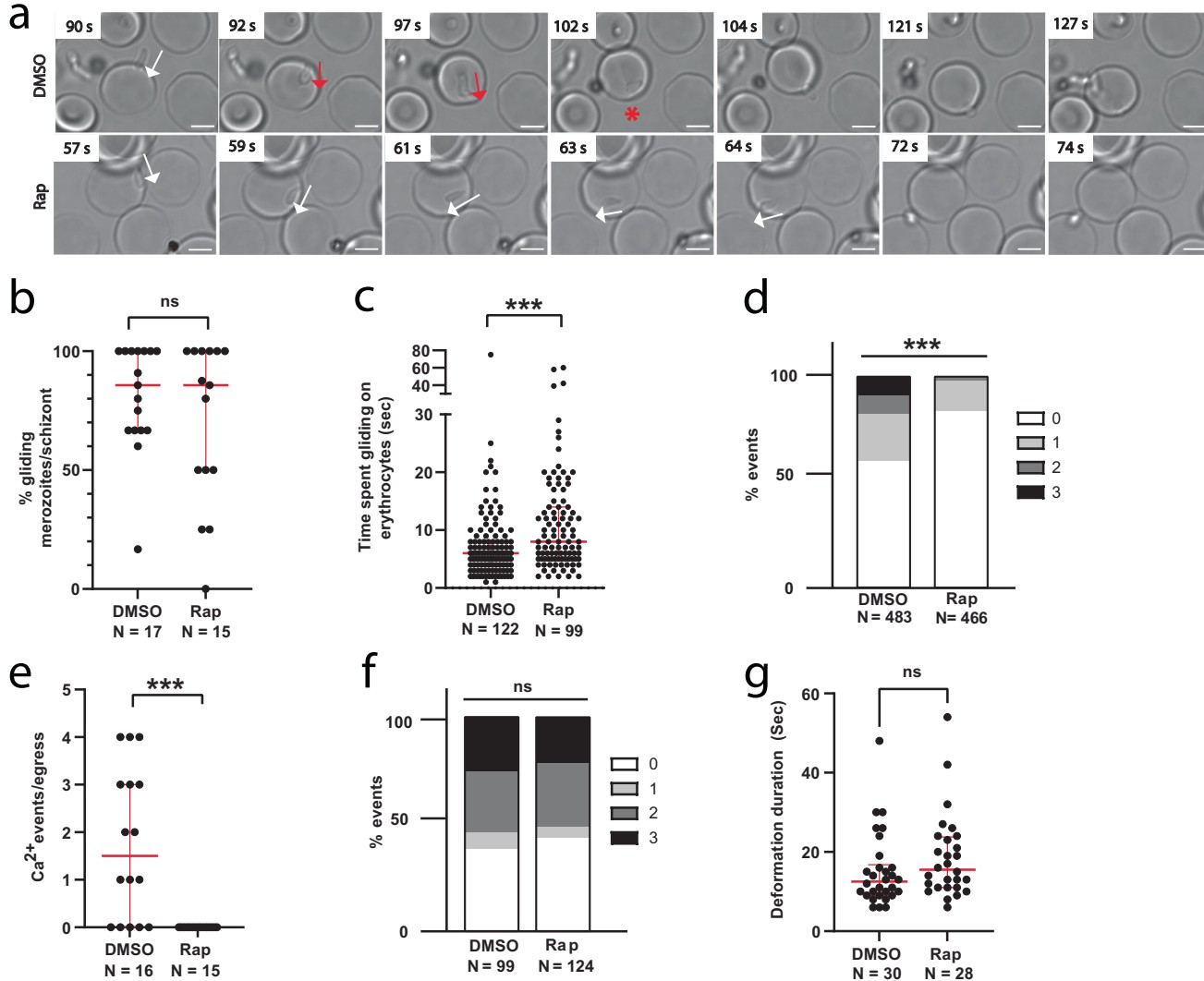

**Fig. 3 | NBPXa is required for human RBC deformation. a** Live stills from Supplementary movie 5 showing DMSO vs. Rap treated NBPXa cKO merozoites interacting with human RBCs. White arrows indicate gliding motility without deformation. Red arrows indicate gliding motility with deformation. Red * = merozoite re-orientating for host cell entry. Scale bars = 2.5 μm. Interactions of DMSO vs. Rap treated NBPXa cKO merozoites with human RBCs were observed to compare (**b**) the % gliding merozoites per schizont (two-tailed Mann-Whitney $U$-test; $p = 0.53$). For DMSO, median = 86% and IQR = 33%; for Rap median = 86% and IQR = 50%. **c** The length of gliding interactions (two-tailed Mann-Whitney $U$-test; $p < 0.0001$) between DMSO (median = 6 s and IQR = 4 s) and Rap (median = 8 s and IQR = 9 s) treated parasites (**d**) the proportion of strength 0-3 merozoite/RBC interaction events (chi-squared test; $p < 0.0001$) and (**e**) the number of Ca²⁺ events seen per egress when merozoites interact with Fluo-4-AM loaded RBCs (two-tailed Mann-Whitney $U$-test; $p < 0.0001$). For DMSO, median = 1.5 events/egress and IQR = 3 events/egress. For Rap, median and IQR = 0 events/egress. Invasion dynamics of the same lines with macaque RBCs were observed to compare (**f**) the proportion of strength 0-3 merozoite/RBC interaction events (chi-squared test; $p = 0.72$) and (**g**) length of time merozoites spent deforming RBCs prior to internalisation (two-tailed Mann-Whitney $U$-test; $p = 0.10$). For DMSO, median = 12.5 s and IQR = 7.75 s. For Rap, median = 15.5 s and IQR = 12.75 s). For all graphs, number of events (*N*) indicated underneath. For all graphs, thick red bars indicate the medians and thinner red bars indicate interquartile ranges. ns = non-significant. Source data for (**b**–**g**) are provided as a Source Data file.

Aside from the two invasions, we also saw no evidence of parasite reorientation. We analysed the interaction of DMSO- and Rap-treated parasites with Fluo-4-AM loaded RBCs and while we observed an average of 1 to 2 Fluo-4-AM signal events per egress for mock-treated parasites (16 schizonts), none was seen for NBPXa-null parasites (15 schizonts; Fig. 3e). Therefore, we conclude that NBPXa is required for deformation of human RBCs and that merozoites lacking this ligand are unable to progress to downstream steps.

In contrast, NBPXa null parasites could deform and invade macaque RBCs as efficiently as controls. There was no significant difference in proportion of strong deformations or the length of time merozoites spent deforming host cells prior to invasion (Fig. 3f, g). These data support that NBPXb is able to compensate for the loss of NBPXa for macaque RBC invasion—strengthening the case that the invasion phenotype observed with human RBCs is directly due to NBPXa deletion, not malfunction of invasion machinery.

## DBPα functions downstream of NBPXa

To investigate the function of DBPα, we began by targeting its host cell receptor (DARC) with an antibody (anti-DARC; Fy6 epitope). Blocking this epitope inhibits both PkDBPα and PvDBP from binding to DARC and thus prevents RBC invasion[18,27,28]. To visualise the effect of blocking DARC binding, we filmed merozoites interacting with human RBCs in the presence of either 5 μg/ml anti-DARC (IC50 = 0.3 μg/ml, Supplementary Fig. 3B), or 5 μg/ml IgG isotype control.

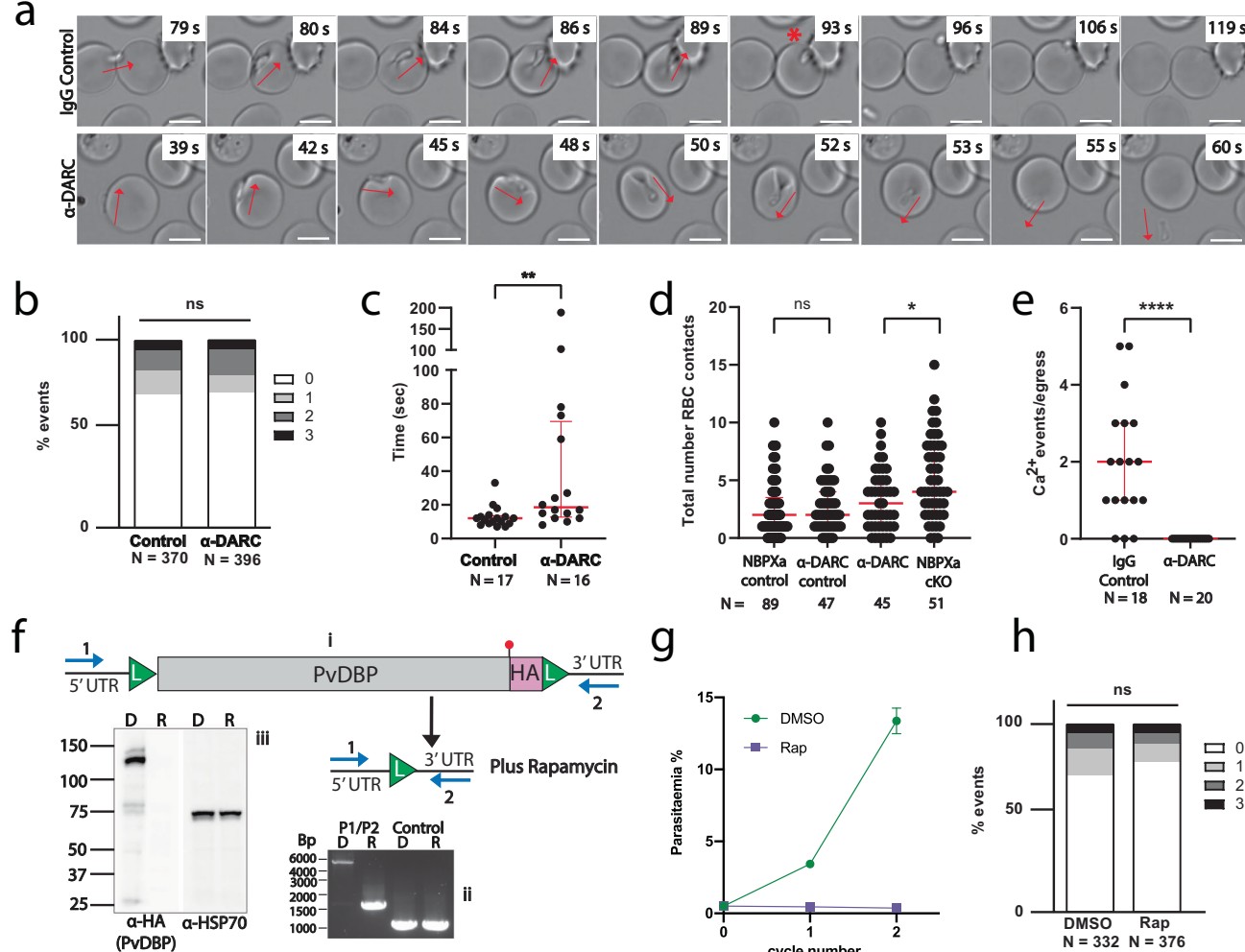

**Fig. 4 | DBPα is required downstream of deformation. a** Panels from Supplementary movie 6 showing merozoite-RBC interactions with 5 µg/ml IgG control (top) or anti-DARC (bottom). Red arrows indicate direction of gliding and deformation. Red * indicates onset of re-orientation and invasion. Scale bars = 5 µm. Invasion dynamics of merozoites with human RBCs in presence of IgG control or anti-DARC were quantified to compare (**b**) the proportion of strength 0-3 merozoite/RBC interaction events (chi-squared test; *p* = 0.19) and (**c**) duration of each score 3 deformation event (two-tailed Mann-Whitney *U*-test; *p* = 0.003). DMSO median = 12 s and IQR = 4.5 s; Rap median = 18.5 s and IQR = 56.75 s. **d** Comparison of total number of RBC contacts made by NBPXa null vs anti-DARC blocked merozoites and controls over 2 min window post egress (two-tailed Mann-Whitney *U*-test; *P* = 0.016 for anti-DARC vs NBPXa null. *P* = 0.22 for NBPXa control vs anti-DARC control. NBPXa control median = 2 s and IQR = 2.5 s; NBPXa null median = 4 s and IQR = 6 s; anti-DARC control median = 2 s and IQR = 3 s; anti-DARC median = 3 s and IQR = 4 s. **e** Number of Ca²⁺ events (per egress) seen when merozoites interact with

Fluo-4-AM loaded RBCs with α-DARC (median and IQR = 0 events) vs IgG control (median = 2 events and IQR = 2 events) (two-tailed Mann-Whitney *U*-test; *p* < 0.0001). **f** Schematic (i) shows excision of *PvDBP* flanked by LoxP sites (L) upon Rap treatment of PvDBP cKO parasites. Diagnostic primer positions noted in blue (P1, P2) produce band shift of 4950 bp (non-excised parasites) to 1670 bp in PCRs (ii), with no change in unrelated control locus. Western blots (iii) probed with HA antibody detect ~120 kDa PvDBP-HA in DMSO (D) but not Rap (R) treated parasites. Loading control = anti-PfHSP70. **g** Growth of PvDBP cKO parasites in human RBCs measured at 24 and 48 h after treatment with either Rap or DMSO. Data mean of 3 independent experiments; error bars +/- SEM. **h** Comparison of strength 0-3 merozoite-RBC interaction events between Rap vs DMSO treated PvDBP cKO parasites (chi-squared test; *p* = 0.11). For all graphs red bars indicated median + IQR, ns = non-significant. Number of events analysed (*N*) indicated under each graph. Source data for (**b**–**h**) are provided as a Source Data file.

Blocking DBPα-DARC interactions completely inhibited invasion (0 invasions, *n* = 20 egresses) compared to control parasites (56 invasions, *n* = 20 egresses). However, gliding and strong host cell deformation were unaffected by blocking DARC (Supplementary movie 6; Fig. 4a, b). This was surprising, as given these merozoites were capable of strong deformation but could not invade, we might have expected them to perform a significantly greater number of strong deformations overall. However, only 3/15 merozoites which generated score 3 events performed an additional score 3 contact (vs. 1/17 for control). Notably, score 3 deformation lasted significantly longer for anti-DARC blocked (median = 18.5 s) vs. control merozoites (median = 12 s; *p* = 0.003; Fig. 4c), suggesting that if strong deformation is initiated but not resolved by reorientation and invasion,

then the merozoite extends this process rather than moving on to another cell. Furthermore, anti-DARC blocked merozoites made fewer RBC contacts overall than NBPXa null parasites, which are not capable of performing strong deformation interactions (Fig. 4d). Thus, strong deformation and in particular, engagement of NBPXa with its host cell receptor, results in a 'semi-committed state' for the merozoite. In such cases, merozoites are unlikely to move to another cell and create a further score 3 interaction, which suggests that this prolonged strong deformation depletes resources required to support subsequent strong interactions (for example reduction of ATP or micronemal protein stores).

DBPα null merozoites have been suggested to be able to reorientate on RBC surfaces[21]. However, we observed no such events for

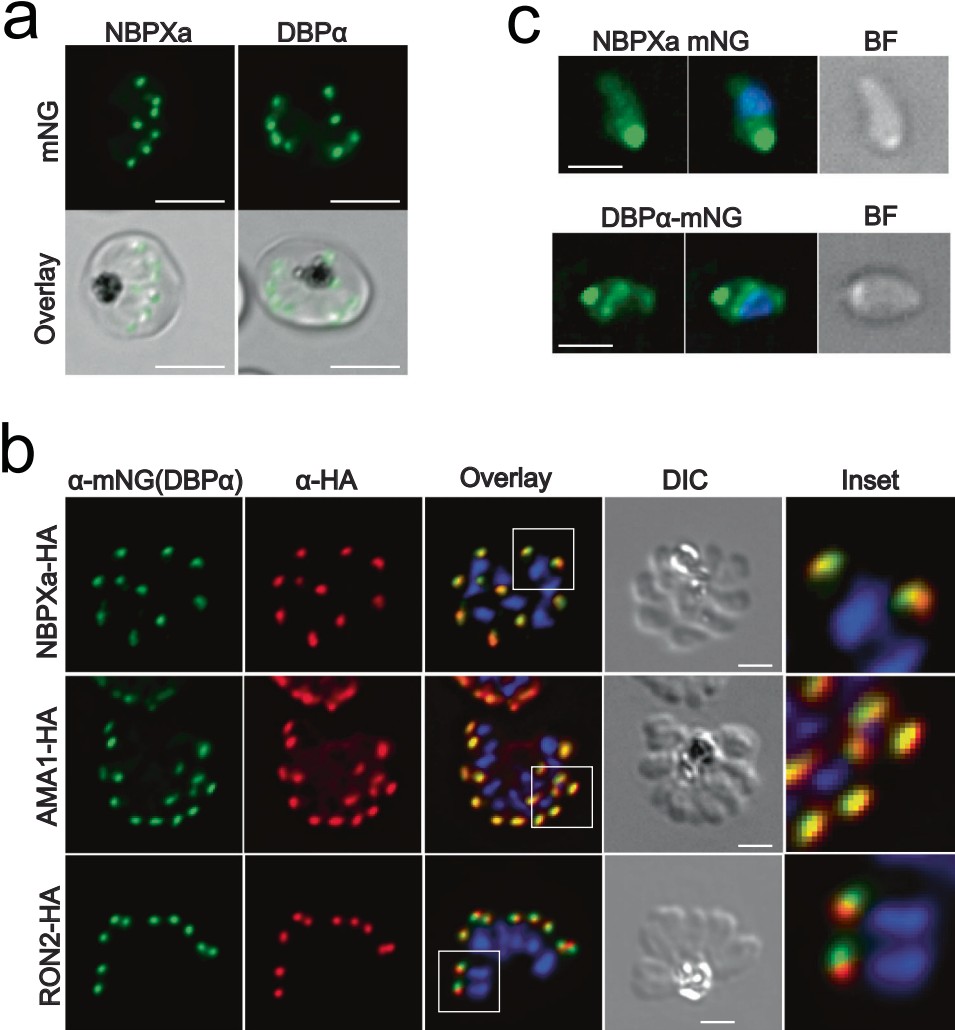

**Fig. 5 | NBPXa and DBPα are closely localised prior to egress but exhibit distinct localisations upon secretion. a** Image of mNeonGreen (mNG) tagged NBPXa and DBPα schizonts. Scale bars = 5 µm. **b** Immunofluorescence assay showing HA tagged NBPXa, AMA-1, and RON2 (red), detected in schizonts using an anti-HA antibody. DBPα-mNG detected in same parasites (green) with an anti-mNG antibody. White boxes depict regions enlarged for 'inset' panels. **c** Extended depth of focus images of NBPXa-mNG and DBPα-mNG in post egress merozoites. Scale bars for (**b**) and (**c**) = 2 µm.

anti-DARC blocked parasites, nor merozoite/RBC fusion events when fluo-4-AM loaded RBCs were treated with anti-DARC (Fig. 4e). In combination, these results suggest that DBPα acts downstream of NBPXa and is required for full commitment prior to re-orientation.

We next sought to generate a DBPα cKO line to compare the phenotype of DBPα null parasites with the effect of anti-DARC treatment. Initial attempts to flox *DBPα* failed due to inappropriate integration events associated with low complexity sequences; therefore, we replaced *PkDBPα* with a floxed *PvDBP* sequence, also expressing an HA tag (Supplementary Fig. 2D), since *PvDBP* fully complements *PkDBPα* in human RBC invasion[18]. Rap treated PvDBP cKO parasites no longer expressed HA-tagged PvDBP (Fig. 4f) and could not proliferate in human RBCs (Fig. 4g). When examined by live microscopy, it was clear that PvDBP null merozoites could glide and deform host cells normally but could not progress beyond this step (Fig. 4h), like wild type parasites in the presence of anti-DARC.

### NBPXa and DBPα exhibit distinct localisations upon secretion from micronemes

Having established sequential roles for NBPXa and DBPα during invasion, we next sought to examine the localisation and secretion dynamics that underpin these roles. We tagged both NBPXa and DBPα

at the C-terminus with the fluorescent marker, mNeonGreen (mNG) (Fig. 5a; Supplementary Fig. 2A, E). We then added an HA tag to either NBPXa, AMA-1, or RON2 in the DBPα-mNG parasite line, and an HA tag to NBPXa in a previously established AMA-1mNG line (Yahata and Hart 2021; Fig. 5b; Supplementary Fig. 2A, C, F)—using iterative CRISPR-Cas9 editing[18].

In schizonts both NBPXa and DBPα localised as a single fluorescent dot at the apical tip (wide end) of each merozoite (Fig. 5a, b). In dual-labelled IFAs DBPα appeared largely colocalised with both NBPXa (Pearson correlation *R* value = 0.94, SD = ± 0.025; *n* = 47 schizonts) and known micronemal marker AMA-1 (Pearson correlation *R* value = 0.96, SD = ± 0.016 *n* = 50 schizonts), confirming that both ligands are micronemal. Rhoptry neck marker, RON2, overlapped slightly with a lower Pearson correlation *R* value, 0.84 (SD = ± 0.035; *n* = 50 schizonts), demonstrating that it resides in separate organelles.

After merozoite egress NBPXa and DBPα had distinct locations (Fig. 5c). Over time, as NBPXa was released from the micronemes, the mNG signal gradually spread evenly across the merozoite (Fig. 6a), until the peak fluorescence intensity of the apical end normalised with that of the merozoite's body (Fig. 6a, cartoon; Supplementary movie 7). In contrast, DBPα exhibited a novel

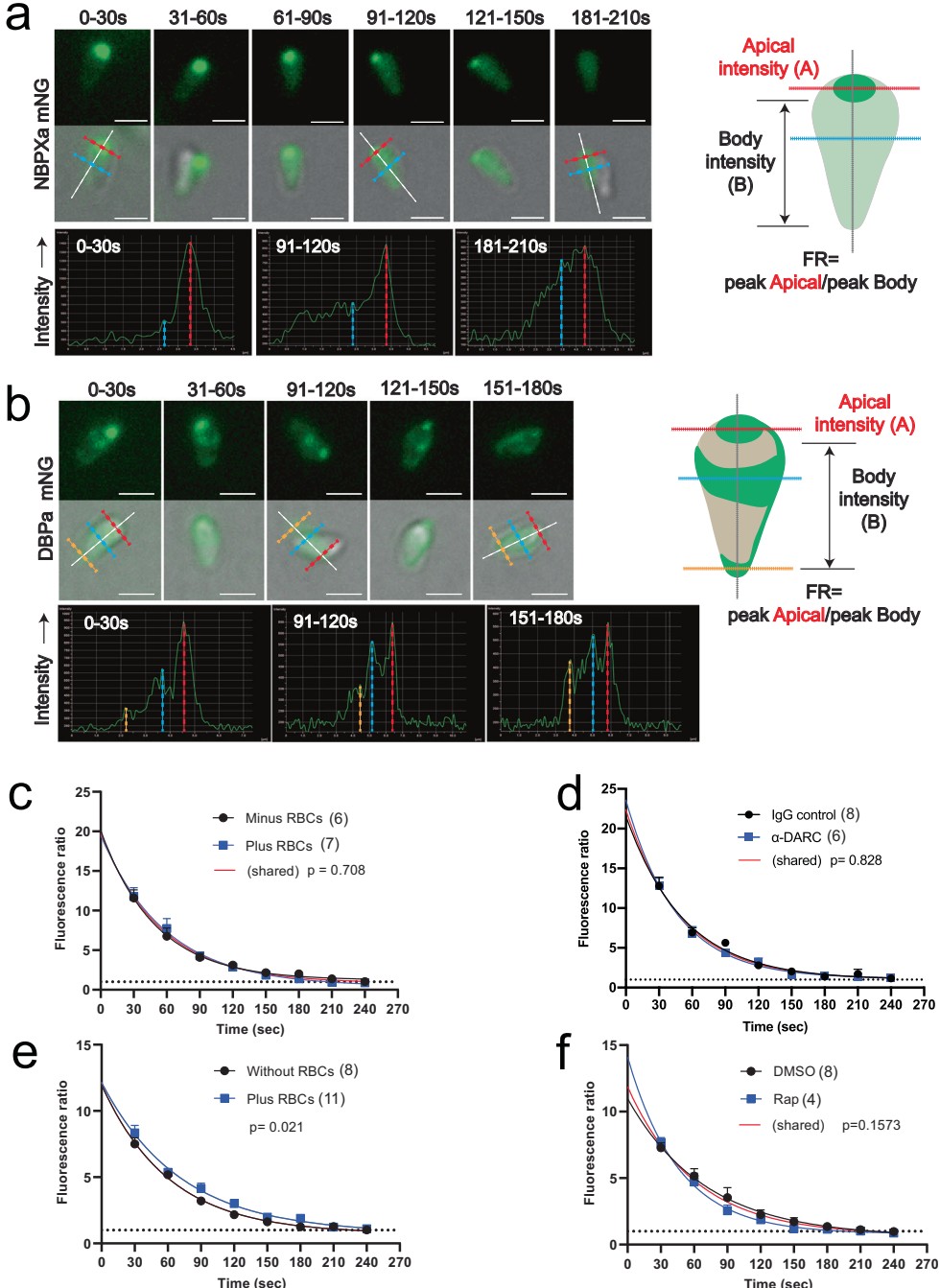

**Fig. 6 | NBPXa and DBPα are secreted gradually and continuously post egress.** Live stills depicting NBPXa (**a**) or DBPα (**b**) secretion over a period of 3 min post egress. Secretion monitored by comparing the peak fluorescence intensity of the merozoite's apical end, to the peak intensity at any point along its body (FR). Merozoite cartoons depict fluorescence pattern of each. NBPXa-mNG (**c**) or DBPα-mNG (**d**) FR values plotted over time +/- RBCs present. **e** NBPXa-mNG secretion with RBCs +/- anti-DARC antibody (**f**) DBPα-mNG secretion with RBCs and +/- NBPXa. For

C-F, measurements were recorded every 30 s after egress (±5 s either side) for all merozoites from a given schizont that were in focus at each timepoint. Each dot represents the average FR value/schizont at a given timepoint. Error bars indicate ±SEM. FR values were fitted using non-linear regression and comparisons between curves were made using an extra sum-of-squares *F*-test; *p* values and number of schizonts (*n*) analysed for each group noted on graphs. Scale bars for (**a**) and (**b**) = 2 µm. Source data for (**c**–**f**) are provided as a Source Data file.

asymmetric surface localisation, accumulating at the widest circumference of the merozoite and the tip of its basal end, resulting in three discrete fluorescence peaks down the length of the merozoite (Figs. 5c and 6b). Interestingly, the localisation had distinct chirality with a solid stripe running down the length of one side of the merozoite (Figs. 5c and 6b, cartoon). When DBPα-mNG tagged merozoites glide, this strip can be seen to rotate in and out of focus (Supplementary movie 8). As secretion of DBPα progressed, the

fluorescence ratio (FR) between the peak apical vs peak body intensity gradually normalised.

## Secretion of RBL/DBP proteins in *Pk* occurs independently of receptor engagement

Conflicting evidence in *Pf* suggests that initial receptor engagement by either RBLs or EBAs is required for secretion of the other RBC protein binding family[12,14]. While in *Pf* RBL and DBP ligands are located in the

rhoptries and micronemes, respectively, *Pk* and *Pv* RBL and DBP homologues are both located in the micronemes[29,30]. Sequential release may still be possible in *Pk* but would require segregation within subpopulations of micronemes. We observed NBPXa and DBPα secretion even without merozoites contacting RBCs (Supplementary movies 7, 8). Therefore, engagement of NBPXa with its host cell receptor is not required for DBPα secretion, and vice versa. We next measured secretion rates of both NBPXa and DBPα under different conditions to explore impact of ligand-receptor binding. NBPXa secretion remained constant, regardless of the presence or absence of RBCs or when DBPα-DARC interactions were blocked with anti-DARC. For all conditions tested, peak apical vs peak body intensity normalised (FR approached 1) within ~240 s after egress (Fig. 6c, d), corresponding to the window of maximum invasion following egress (Supplementary Fig. 1H). Likewise, DBPα secretion progressed rapidly over a similar window. A slight, but significant decrease in DBPα secretion speed was observed in the absence of RBCs (Fig. 6e). However, DBPα secretion remained constant when RBCs were present, but NBPXa was conditionally deleted (Fig. 6f). Therefore, our data demonstrate that micronemal secretion of RBL/DBPs is not dependent on receptor engagement for either NBPXa or DBPα in *Pk*.

### DBPα co-localises with the early moving junction before host cell entry

Several *Pf* DBL and RBL ligands localise to the moving junction of merozoites when treated with cytochalasin D (cyto D) to block internalisation, including PfEBA-175 and PfRh1[13,31,32]. However, in our NBPXa mNG parasite line, the location of NBPXa did not change during merozoite internalisation. Apical NBPXa, likely representing un-secreted micronemal stores, moved into the RBC and did not form a junction-like structure (Supplementary movie 9). Comparing the location of NBPXa-HA and AMA1-mNG by IFA, we found that cyto D-blocked merozoites had a single apical NBPXa signal, moving through the 'double dot' pattern seen from a cross section of AMA-1 labelled moving junction (Fig. 7a).

We tagged NBPXa at its C-terminus, which is proteolytically cleaved away from the ectodomain at some point during invasion. Attempts to localise the NBPXa ectodomain by IFA with a rabbit polyclonal antibody raised against the putative NBPXa RBC-binding domain (amino acids 151-467; Fig. 6b, schematic) were unsuccessful, as our antibody was not compatible with IFA conditions that permit differentiation between surface vs intracellular localisation. However, western blots revealed that most of the NBPXa ectodomain was present in the culture supernatant as a single product following cleavage from its HA-tagged cytoplasmic tail (Fig. 7b). Consistent with imaging results, processing was unaffected by blocking invasion with anti-DARC antibody (Fig. 7c). This suggests that NBPXa is continuously secreted following egress and cleaved from the merozoite surface, potentially at the putative transmembrane ROM4 cleavage site, like its *Pf* counterparts[31,33] (Fig. 7d).

DBPα-mNG largely co-localised with HA-tagged AMA-1 before host cell entry, with a circumferential 'double dot' structure at the merozoite-RBC interface characteristic of the early moving junction (Fig. 7a). However, live-cell imaging showed that the DBPα double dot pattern remained at the apex of invading parasites (See Supplementary movie 10, at 23 s onwards) rather than travelling backwards along the parasite-RBC interface during entry—as is seen for typical moving junction markers, such as AMA-1. Thus, DBPα may play a role at or just before formation of the moving junction but is not part of the moving junction during entry.

### Simultaneous targeting of PkRBL and PkDBP invasion pathways inhibits invasion synergistically

Having established sequential roles for PkRBL/DBP family proteins during invasion, we sought to determine whether targeting both

pathways would result in synergistic invasion inhibition. Rabbit poly-clonal antibody targeting the putative RBC binding domain of NBPXa showed dose-dependent invasion inhibition of *Pk* with an IC$_{50}$ of 4.9 mg/ml (Fig. 7e), not dissimilar to inhibition by polyclonal anti-PvDBP RII antibodies[34]. To determine whether combining NBPXa and DBP antibodies would potentiate inhibition, we used our *Pk* PvDBP$^{OR}$ line—combining our NBPXa polyclonal antibodies with DB10, a well characterised human monoclonal against PvDBP-RII[34]. Combining a fixed quantity of DB10 with a titration of NBPXa antibodies, we observed an increase in IC$_{50}$ relative to the calculated Bliss Additivity curve—demonstrating clear synergistic activity between DBP and NBPXa antibodies (Fig. 7e, f).

## Discussion

The tractability of *Pk* for genetic manipulation, together with the relatively large size of its merozoites have enabled us to delineate the morphological and molecular processes underlying invasion. Our results show that gliding motility, deformation, merozoite-RBC fusion event, and re-orientation, are distinct, essential steps leading to host cell entry. We have also shown for the first time that although the *Pk* RBL and DBP ligands may be secreted simultaneously, following secretion and during invasion they exhibit distinct localisations. Our cKO data also demonstrates NBPXa and DBPα have distinct roles, with DBPα functioning downstream of NBPXa.

NBPXa null merozoites retain their ability to glide but cannot deform human RBCs, demonstrating that both gliding motility and NBPXa-receptor interactions are required for deformation. Yet how does NBPXa mediate deformation? NBPXa may be coupled to the merozoite's actomyosin motor, as suggested but not confirmed for the *Pf* RBL/DBPs[35,36]. Alternatively, NBPXa-receptor binding, in combination with the rotational movement of gliding motility[22], may pull the RBC membrane around the merozoite. Enzymatic cleavage of NBPXa at its putative rhomboid ROM4 site may sever these links, allowing the merozoite to continue its forward trajectory. ROM4 processing of *Pf* RBL and DBP ligands is predicted to occur during host cell entry, to shed the ecto-domains of these proteins during internalisation[31–33,37]. However, clea-vage of these ligands occurs whether or not *Pf* merozoites re-invade[37] in line with what we observe for NBPXa. Thus, rhomboid cleavage of RBL ligands may primarily facilitate deformation instead, and the role of the RBLs may be to increase the "stickiness" of the parasite to create torsion-driven wrapping. Future work using double-tagged lines expressing, for instance, a fluorescent marker at the N-terminus of NBPXa in addition to its C-terminus, may yield real-time results that explain the secretion and processing dynamics of RBL ligands more definitively.

We also demonstrate that blocking DBPα-DARC interactions prevents merozoite apical fusion with the host cell and reorientation on the RBC surface. Interestingly, this mimics the consequence of disrupting the *Pf* Rh5/CyRPA/Ripr complex[4,11], though *Pk* lacks an PfRh5 ortholog. However, *Pk* does have orthologs of PfCyRPA and PfRipr, which are two essential proteins[26,38–40], Further work will be required to dissect the molecular steps leading to merozoite reor-ientation and to determine the function of DBPα relative to that of the PkRipr-complex.

The implications for understanding *Pf* invasion are more complex, with previous evidence suggesting overlapping functions for that DBP/EBA and RBL/RH ligands. Improved deformation may enable a lower affinity EBA interaction to be successful and vice versa. It is also plausible that *Pf* uses these families in a more interchangeable way, underpinned by some key differences such as duplicated RBC-binding domains in EBA175 (vs single domain in DBPs), the distinct location and secretion timings of the PfRH and PfEBA proteins[41]. Nevertheless, visualising the key steps of invasion which are difficult to display in *Pf* provides a new perspective on key events shared across the genus—most notably that deformation and reorientation are clear and sepa-rate events, and that a merozoite-host cell fusion event and potentially

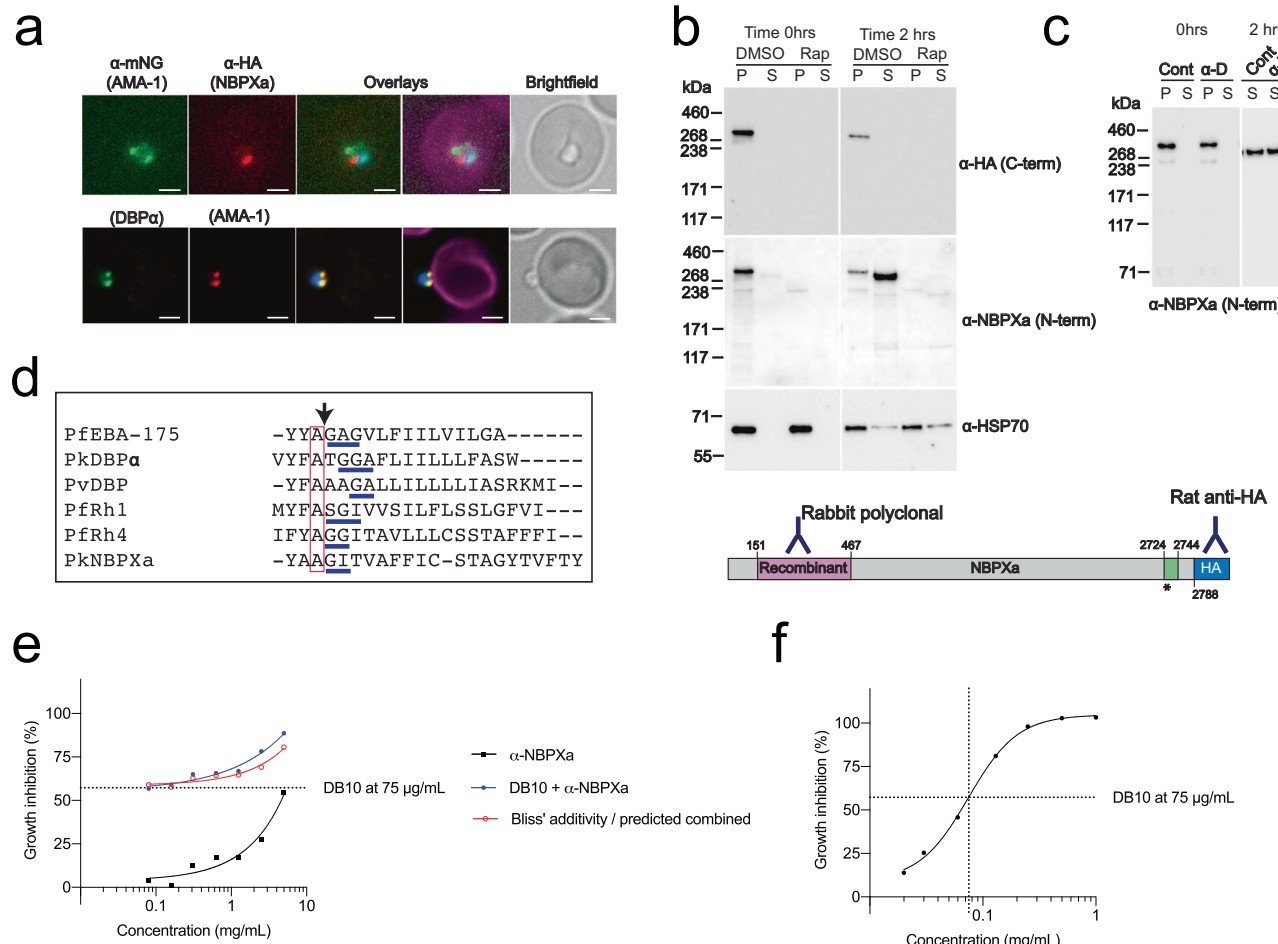

**Fig. 7 | NBPXa acts prior to moving junction formation and can be synergistically targeted with DBP. a** IFAs of *Pk* merozoites stalled mid-invasion with cyto D. Top panels depict AMA1-mNG + NBPXa-HA. Bottom panels depict DBPα-mNG + AMA1-HA. Overlays stained with wheatgerm agglutinin (Magenta). Scale bars = 2 μm. **b** Processed NBPXa is detected by western blot in culture supernatants (S) when egressing merozoites are allowed to re-invade fresh RBCs and (**c**) when invasion is blocked with anti-DARC (α-D). Cont = control IgG. For all blots, NBPXa was detected with rat anti-HA (targeting C-term) and/or rabbit anti-NBPXa raised against the NBPXa N-terminus (amino acids 151-467). Anti-PfHSP70 used as loading control. P = pellet fraction. Green region in schematic shows transmembrane domain, * indicates putative ROM4 cleavage site. **d** Clustal alignment of several predicted RBL/DBP transmembrane domains. All contain a conserved alanine residue (red box), followed by putative helix destabilising motifs underlined in blue, which may serve as ROM4 recognition sites. All sequences apart from NBPXa from Baker et al.[33] & O'Donnell et al.[31]. The experimentally determined PfEBA-175 cleavage position is indicated with black arrow. (O'Donnell et al.[31]). Parasite growth inhibition activity of anti-NBPXa (**e**) and anti-DB10 (**f**) was assessed individually and in combination (**e**) using fixed DB10 concentration with increasing anti-NBPXa concentrations Results calculated from two independent experiments. Error bars indicate ±SEM. Source data for (**b**), (**c**), (**e**) and (**f**) are provided as a Source Data file.

tight junction formation occur before reorientation. Cross-species comparisons of invasion provide an invaluable tool to understand the biology underpinning these complex interactions and how to target them across all *Plasmodium* species.

Finally, our results reveal a mechanism for a stepwise commitment to invasion, which may underpin *Plasmodium* species host cell tropism (Fig. 8). For all species, RBL and DBP repertoires determine the range of RBCs amenable to invasion[8]. Most *Pf* RBL/DBP receptors are broadly distributed across human RBCs of different age and blood type, enabling this species to infect a wide range of individuals[8]. However, some non-Laveranian species are restricted to growth in RBC subsets including *Pv* and *P. ovale*, to reticulocytes[42] and *Pk* and *Pv*, in most instances, to Duffy positive RBCs.

Yet, how do merozoites ensure they detect, bind to, and invade rare RBC subsets without depleting valuable resources attempting to invade unsuitable host cells? Our data indicate that NBPXa, and by extension, essential RBLs of other non-Laveranian species, mediate the first phase of commitment to invasion: host cell deformation. NBPXa null *Pk* merozoites do not deform RBCs on contact, and subsequently make a greater number of RBC contacts overall. In contrast, when

NBPXa is present but DBP-receptor interactions are blocked, merozoites spend more time interacting with RBCs they deform but cannot invade, reducing their chances of subsequent successful interactions (Fig. 8). In vivo, this property of RBLs may present the parasite with a distinct advantage: the ability to sample host cells efficiently by quickly gliding over RBCs lacking an appropriate RBL receptor and thereby saving finite resources for invasion of an appropriate receptor-positive cell. This may explain how *Pv* merozoites can efficiently invade reticulocytes (which form ~1% of RBCs in peripheral blood), as the reticulocyte-specific RBL proteins ensure the merozoite is "blind" to normocytes. Testing this hypothesis will be challenging due to our inability to culture *Pv* in vitro but combining orthologous gene replacement approaches in *Pk* and *P. cynomolgi* and using *Pv* in ex vivo experiments, could offer routes to address this.

The proposed sequential, phased commitment to invasion presents key opportunities for targeted vaccine design. In recent years, work to target *Pf* RBL and DBP proteins (apart from PfRh5) as vaccine candidates has been hampered, by both redundancy and polymorphism[43,44]. Our work shows that this is not the case for *Pk*. The growth inhibition assays with anti-PvDBP and anti-PkNBPXa antibodies

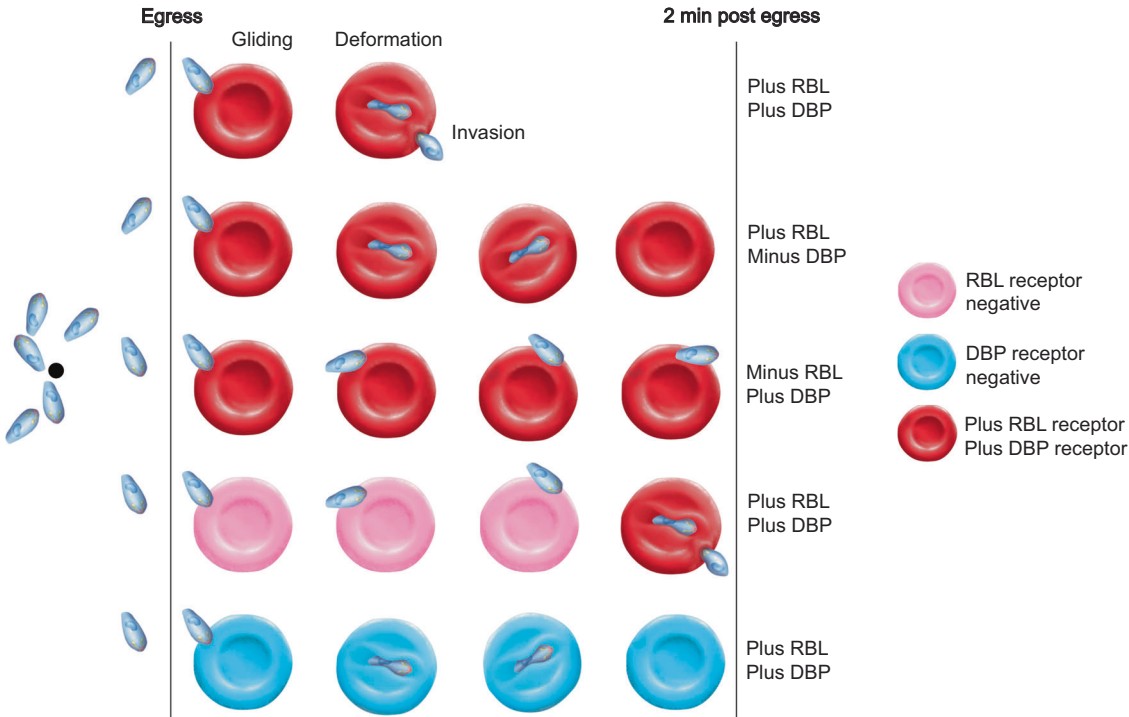

**Fig. 8 | A model of how RBL and DBP ligands mediate phased commitment to invasion.** Both in the absence of NBPXa and when host cells lack appropriate RBL receptor(s), *Pk* merozoites contact a greater number of RBCs overall, increasing their chances of interacting with a suitable RBC. In contrast, DBP null merozoites, or WT parasites contacting Duffy negative host cells, form semi-committed interactions, thereby reducing their chances of contacting a suitable host cell.

demonstrated that blocking two critical, sequential invasion steps in combination dramatically curbs parasite growth compared to targeting either ligand individually. These results may be translatable to other *Plasmodium* species. Targeting a critical PvRBL ligand[45] together with PvDBP may be the key to developing an effective *Pv* vaccine, building upon promising results from recent PvDBP Phase II clinical trials[46].

## Methods

### Recombinant NBPXa production

A gene fragment of PkNBPXa encoding amino acids 151 to 467 (IDRILD... to ...DALKDK) were flanked by unique NotI and AscI restriction enzyme sites and cloned into a pTT3-based mammalian expression plasmid between a mouse antibody N-terminal signal peptide and a C-terminal tag that included a protein sequence that could be enzymatically biotinylated by the BirA biotin ligase and 6-His tag for purification [PMID: 24620899]. The ectodomains were expressed as soluble recombinant proteins in HEK293 cells as previously described[47]. To prepare purified proteins for immunization spent culture medium containing the secreted ectodomain was collected from transfected cells, filtered and purified by $Ni^{2+}$-NTA chromatography using HisTRAP columns using an AKTAPure instrument (GEHealthcare). Proteins were eluted in 400 mM imidazole as previously described [PMID: 31952523], and extensively dialysed into HEPES-buffered saline (HBS) before being quantified by spectrophotometry at 280 nm. Protein purity was determined by resolving purified protein by SDS–PAGE using NuPAGE 4–12% Bis Tris precast gels (ThermoFisher) at 200 V. Purified proteins were aliquoted and stored frozen at −20 °C until used. Where enzymatically monobiotinylated proteins were required to determine antibody titres by ELISA, proteins were co-transfected with a secreted version of the protein biotin ligase (BirA) as previously described [PMID: 27226583], and extensively dialysed against HBS and their level of expression determined by ELISA using a mouse monoclonal anti-His antibody (1/1000; His-Tag monoclonal antibody, 70796, EMD Millipore) as

primary antibody and a goat anti-mouse alkaline phosphatase-conjugated secondary (1/5000; A3562, Sigma-Aldrich).

### NBPXa antibody generation and purification

Rabbit anti-PkNBPXa antibodies were raised (Covalab) against the recombinant protein described above and subsequently purified with an immunoaffinity column prepared by coupling recombinant protein to 1 ml of activated sepharose beads. In brief, anti-NBPXa antibodies were purified by diluting equal parts rabbit serum and 10 mM Tris (pH 7.5) and running over column 3 times. The column was then washed by addition of 20 mL 10 mM Tris (pH 7.5), then again with 20 mL 10 mM Tris (pH 7.5) with 0.5 mM NaCl. Bound anti-PkNBPXa was eluted by passing 10 mL glycine over the column and collecting in a tube containing 700 μL 1 M Tris HCl (pH 8.0) to neutralise the glycine. Eluted protein was concentrated and buffer exchanged into incomplete media (ICM) using the Amicon® Pro Purification System with a 50,000 molecular weight cut-off (MWCO), as per the manufacturer's instructions. Final antibody concentration was read using the DeNovix DS-11 Series microvolume spectrophotometer, blanked with ICM and with the antibody mass extinction coefficient (E1%) set to 14.4 for rabbit IgG. Concentrated, buffer exchanged IgG was stored at −20 °C until use.

### Generation of DNA constructs for transfection

**Gene IDs.** *P. knowlesi* normocyte binding protein Xa (*NBPXa*), Duffy binding protein alpha (*PkDBPα*), Apical membrane antigen 1 (*AMA-1*), and Rhoptry neck protein 2 (*RON2*) constructs were derived from *P. knowlesi* strain H sequences from the PlasmoDB Database (www.plasmodb.org) with respective accession numbers: PKNH_1472300, PKNH_0623500, PKNH_0931500, and PKNH_1230100.

**Cas9 guide plasmids.** All Cas9 guide RNA sequences (NBPXa N-term: AAATTCATGAACCCCAATTA; RON2 C-term: ACGCCCGCATACAGATGTA; and DBPa-C-term: CATGCAGCAGTTCACCCCCC), apart from one targeting the NBPXa C-terminus (GTAACGAATATATATGAGTA), were

inserted into vector pCas9/sg according to previously published methods (Mohring et al.[18]). Because of difficulty cloning the NBPXa C-term guide into pCas9/sg, this sequence was inserted into a second Cas9 vector, pCas9HF/sg. This vector contains all component of pCas9/sg but has a smaller backbone and a high fidelity (HF) Cas9 sequence instead of the wild type version. Insertion of the NBPXa C-term guide into pCas9HF/sg was achieved by amplifying the entire guide cassette using overlapping primers (Supplementary Tables 1, 2) containing the guide sequence. This PCR product was digested with PspXI and BstEII (NEB) and ligated into the pCas9HF/sg using T4 DNA ligase (Promega). Previously published guide plasmids pCas9_p230p[18] and pCas9_AMA1_Cterm[22] were used to target the p230p and AMA-1 loci, respectively.

### Donor DNA constructs

**Generation of DiCre cassette donor DNA**. The donor plasmid for introducing the DiCre cassette into Pkp230p was generated in multiple steps. Primers and templates used to amplify each component are listed in Supplementary Tables 1, 2. First, 5′ and 3′ 400 bp homology regions flanking the Pkp230p guide were amplified from Pk A1-H.1 genomic DNA and inserted into a plasmid backbone containing a multiple cloning site (synthesized by Geneart, Thermofisher) via SacII/SpeI and BlpI/NcoI restriction sites, respectively.

To generate a Cre60 cassette, the PkHSP70 5′UTR sequence was amplified and inserted into a pGem-T Easy vector (Promega) followed by the Cre60 sequence via restriction cloning between XhoI/KpnI sites, and the PfPbDT 3′UTR sequence between KpnI/BlpI sites.

Likewise, to generate a Cre59 cassette, the PkEF1α 5′UTR sequence was amplified and inserted into the pGem, followed by the Cre59 sequence (XhoI/KpnI) and PfHRP2 3′UTR sequence (KpnI/SpeI). Finally, the PkHSP70 5′UTR-Cre60-PbDT 3′UTR cassette was introduced between p230p homology arms via SpeI/BlpI restriction sites, and the PkEF1α 5′UTR-Cre59-PfHRP2 3′UTR cassette via SpeI/NotI restriction sites.

**Generation of PkNBPXa, PkDBPα, PkRON2, PkAMA-1, and PvDBP tagging and cKO donor DNA constructs**. Donor DNA templates used to modify PkDBPα, PkNBPXa, PkRON2, PkAMA-1, and PvDBP with either tags or LoxP sites were generated by overlapping PCR[18] using primers and templates listed in Supplementary Tables 1, 2. The NBPXa-mNG, NBPXa-HA, RON2-HA, and AMA1-HA constructs were transfected as PCR products. Prior to transfection, the DBPα-mNG, NBPXa-NtermLoxP, and NBPXa-CtermLoxP/HA constructs were first cloned into plasmid backbone (Geneart, Thermofisher Scientific), containing a multi-cloning site with SacII, NotI, and EcoR1 restriction sites. The final PCR amplification step of the DBPα-mNG construct added a SacII restriction site to the 5′ end of the construct and an EcoR1 site to its 3′ end for insertion into vector backbone. Likewise, the final PCR amplification step of NBPXa-NtermLoxP and NBPXa-CtermLoxP/HA constructs added a SacII restriction site to the 5′ ends and a NotI restriction site to the 3′ ends of both constructs for introduction into PkB1 by restriction cloning.

In order to generate the PvDBP cKO donor template, plasmid pDonor_PvDBP^OR [34] was sequentially modified to introduce LoxP sites flanking the PvDBP sequence. The plasmid was first linearized with BstBI and NotI to remove 90 bp at the C-term end of PvDBP^OR. This 90 bp was replaced with an insert including a HA tag-LoxP sequence. A second LoxP sequence was introduced before the PvDBP start codon by removing homology region 1 (HR1) via SacII/SpeI and replacing it with a HR1-LoxP sequence.

### In vitro maintenance and synchronization of *P. knowlesi* parasites

Blood stage A1-H.1 P. knowlesi parasites were cultured in human erythrocytes (UK National Blood Transfusion Service) at a 2% haematocrit with custom made RPMI-1640 medium, supplemented with 10% Horse Serum (v/v) and 0.292 grams/litre (2 mM) L-glutamine according to

previously established methods[48]. Parasites were tightly synchronized for experiments by first purifying schizonts with a density gradient centrifugation step using 55% Nycodenz (Axis-Shield) and allowing parasites to re-invade fresh RBCs over a 1–2 h window. Next, newly formed ring stages were isolated by incubation with 140 mM guanidine hydrochloride for 10 min to kill all parasites older than 5 h old, according to a previously established protocol[49].

### Transfection and genotyping of *P. knowlesi* parasites

Transfections were performed with 10–20 μl mature schizonts, 10 μg of the Cas9 plasmid, and 20 μg of the donor DNA using the Amaxa 4-D Nucleofactor X machine (Pulse code FP158) and P3 Primary Cell Kit (Lonza)[18,48]. Transfected parasites were cultured with 100 nM pyrimethamine for 5 days following transfection in order to select for parasites that had taken up the Cas9 plasmid. A minimum of 1 week later, cultures were subsequently treated with 1 μM 5-Fluorocytosine for 7 days to eliminate parasites still harbouring the Cas9 plasmid. Finally, transgenic parasites were cloned by limiting dilution.

Transfected parasites were analyzed by diagnostic PCR using GoTaq Green master mix (Promega) and the following conditions: 3 min at 96 °C, then 30 cycles of 25 s at 96 °C, 25 s at 55 °C, and 1 min/kb at 64 C. Diagnostic primers along with expected PCR product sizes are listed in Supplementary Tables 1, 2.

### Rapamycin treatment of conditional knockout lines

Tightly synchronised ring stage P. knowlesi cultures were split in two and incubated in complete media containing 10 nM rapamycin (Sigma-Aldrich) or the equivalent volume of its carrier, DMSO (0.005%). Parasites were treated for a minimum of 3 h at 37 °C, before drug or DMSO containing media was removed and cultures were re-suspended in fresh, drug-free complete media.

Diagnostic PCRs were carried out using GoTaq Green to detect excised or non-excised parasites after drug or mock treatment. For PkNBPXa cKO parasites, primers Fwd-GTTTCACTATTGAAGGATAATTTTAGGAAAGG-3′ and rev-GTATTACGGTTAATATGTTTTAACGTAACTGG were used to detect excised parasites, giving an expected product size of 606 bp. Primer pair Fwd-ACTGCCGGATACACAGTCTTTAC and Rev-GTATTACGGTTAATATGTTTTAACGTAACTGG were used to detect non-excised parasites, with an expected product size of 404 bp. For PvDBP cKO parasites, primers Fwd- CCATGTACACGATTTGTGTACTTATAGAATC and Rev- GTAGGGAACATTTCTTTCTGCGG were used to detect both WT (expected product size = 4950 bp) and excised (expected product size = 1670 bp) parasites.

A positive control using primers Fwd-CCCGGGGCGTTTTCGCGTATCTGCGCTTTTTC and Rev-CCTAGGGGACAATATATCCTCACAGAACAACTTG, which amplified a 1043 bp product from the PkMTIP locus, was also included in each set of reactions.

### Parasite multiplication assays

To determine the multiplication rate of mutant parasites, ring stage DMSO and rapamycin-treated cultures were adjusted to a 0.5% parasitaemia and 2% haematocrit and were grown in triplicate in 96 well plates in a gassed chamber at 37 °C. After 24 h, a starting sample was taken and stained with SYBR Green I (Life Technologies) before measuring parasitaemia by flow cytometry (FACS). A second sample was taken 26 h later, and a final sample was taken roughly 26 h later again. Data were collected on an Attune NxT flow cytometer using FACSDiva 6.1.3 software. Data were analysed using FlowJo v10. Graphpad Prism v10 was used for statistical analysis. Representative gating strategies are outlined in Supplementary Fig. 4.

To determine the invasion inhibitory potential of anti-DARC (2C3 clone, Absolute Antibody), anti-DB10[34] and anti-NBPXa antibodies, growth inhibition assays (GIA) were carried out using an LDH assay as previously described[50]. Bliss's additivity was calculated using a previously described equation[51].

## Immunofluorescence assay (IFA) analysis

Late stage schizonts were thinly smeared on glass slides, air dried, and fixed in 4% paraformaldehyde (PFA) in PBS for 30 min, followed by 0.1% Triton X-100 in PBS for 10 min. Slides were subsequently blocked in 3% BSA in PBS overnight. All primary and secondary antibodies were diluted in blocking solution and incubated sequentially for 1 hr at room temperature followed by three washes in PBS for 10 min each. Slides were incubated with the following antibodies: mouse α-mNeonGreen (1/500; 32F6, Chromotek) followed by donkey anti-mouse IgG (H + L) Highly Cross-Adsorbed Secondary Antibody, Alexa Fluor™ 488 (1/1000, Invitrogen); rat α-HA (1/500; 3F10, Sigma), followed by IRDye® 680RD Goat anti-Rat IgG Secondary Antibody (1/1000; LI-COR Odyssey); and rabbit anti-PkMSP1[20] (1/2000) followed by goat anti-rabbit IgG (H + L) Highly Cross-Adsorbed Secondary Antibody, Alexa Fluor™ 488 (1/1000, Invitrogen). After a final wash in PBS (3 × 10 min), slides were mounted in ProLong Gold Antifade Mountant containing DAPI (Thermo Fisher Scientific). Co-localisation of dual-labelled proteins was assessed by Pearson's correlation using the NIS Advanced Research software colocalization module.

For in solution IFAs, merozoites were captured mid-invasion by treating egressing cultures with 100 nM cytochalasin D (Sigma) for 30 min and then gently spinning cultures down at 1500 × g for 1 min. Parasite pellets were re-suspended in a 4% PFA/0.0025% glutaraldehyde solution in PBS, and then loaded into a poly-L-lysine coated μ-Slide VI 0.4 (Ibidi). Parasites were incubated in fixative solution for 30 min before fixative was removed by pipetting the solution out of the channel slides.

Channels were then washed 5× by gently pipetting PBS into the channel and then removing it in the same manner as the fixative. Subsequently, parasites were incubated with 0.1% Triton X-100 in PBS for 10 min, followed by 5 wash steps in PBS. Washed parasites were blocked with 3% BSA in PBS for at least 1 h. Samples were subsequently incubated with primary and secondary antibodies as described above, with 5× wash steps in between each incubation. After the final wash steps, parasites were incubated in PBS containing Hoechst 33342 (Cell Signaling Technology) for half an hour prior to imaging. All samples were imaged with a Nikon Ti E inverted microscope using a 100× or 60 × oil immersion objective and an ORCA Flash 4.0 CMOS camera (Hamamatsu). Images were acquired, processed and statistically analysed using the NIS Advanced Research software package.

## Immunoblot analysis

Saponin released schizonts were lysed in four volumes of a CoIP buffer containing: 10 mM Tris/Cl pH 7.5, 150 mM NaCl, 0.5 mM EDTA, 0.5% NP-40, 1 mM PMSF, and 2× cOmplete EDTA free protease inhibitors (Roche). Lysates were incubated on ice for 10 min and then centrifuged at 12,000 × g for 20 min to remove residual solid material. Subsequently, 1 volume of 2× LDS sample buffer (Invitrogen) was added to the soluble fraction. For NBPXa processing assays, pellets were prepared as described above, and culture supernatants were first concentrated 5× in Vivaspin 500 concentrators (Sartorius) before adding LDS sample buffer. All samples were boiled for 5 min prior to separation by SDS PAGE on precast 3–8% Tris-Acetate NuPAGE gels (Invitrogen). SDS-PAGE fractionated proteins were transferred to a nitrocellulose membrane using a semidry Trans-Blot Turbo Transfer System (Bio-rad). Membranes were blocked in 10% (w/v) milk in 0.1 % PBS-Tween-20 (PBST) overnight. All primary and secondary antibodies were diluted in 1% (w/v) skimmed milk in PBST and incubated sequentially for 1 hr at room temperature followed by three washes in PBST for 10 min each. The following antibodies were used: rat α-HA (1/5000; 3F10, Sigma) and rat α-PfHSP70[52] (1/2000), followed by Goat anti-rat IgG-HRP (1/5000, Bio-Rad), and rabbit α-PkNBPXa (1/5000), followed by Goat anti-rabbit IgG-HRP (1/5000, Bio-Rad). After the final washes, membranes were incubated with Clarity Western ECL substrate (Bio-Rad) and developed using a Chemidoc imaging system (Bio-Rad).

## Live cell imaging

Purified schizonts were added to fresh human or macaque RBCs to make a 10–15% parasitaemia and 2.5% haematocrit culture. The haematocrit was subsequently adjusted to 0.25% in complete media, and 150 μl was loaded into a poly-L-lysine coated μ-Slide VI 0.4 (Ibidi). For anti-DARC assays, either 5 μg/mL human anti-DARC (2C3clone, Absolute Antibody), or 5 μg/mL anti-human IgG (Invitrogen) was first added to the 0.25% haematocrit culture, prior to loading samples into the channel slides. For calcium flux assays, RBCs were first incubated with 5 μM fluo-4-AM (Invitrogen) in RPMI for 1 h at 37 °C. Cells were subsequently washed three times in RPMI, and then allowed to rest at 37 °C for a further half an hour for de-esterification to occur, prior to mixing with parasites. For secretion assays performed in the absence of fresh RBCs, late stage schizonts were purified and diluted sufficiently in complete media so that egressing parasites could not contact surrounding parasitized RBCs. Loaded slides were subsequently transferred to a Nikon Ti E inverted microscope chamber, pre-warmed to 37 °C. Samples were imaged using a 100× or 60 × oil immersion objective and an ORCA Flash 4.0 CMOS camera (Hamamatsu), at a rate of 1 frame/sec (100 msec exposure for each channel). Videos were acquired and processed using the NIS Advanced Research software package. Fluorescence intensities were measured using the intensity profile feature of the NIS Advanced Research software package.

## Statistics and reproducibility

For live cell imaging experiments, at least 2–3 independent repeats were performed per condition, and typically, events resulting from at least 10–20 egresses were analysed. Specific sample sizes (number of egresses or merozoite-RBC interactions) are indicated under each graph/within figure legends. For quantification of parasite replication by flow cytometry, 3–5 independent experiments, each with 3 technical repeats, were performed. For quantification of parasite growth by LDH assay, 2–3 independent experiments were performed. Westerns, PCRs, and IFAs demonstrating PkNBPXa and PvDBPa cKO status were repeated independently 3 times. For PkNBPXa and PkDBPα secretion assays, 2 biological repeats were performed. All statistical analysis was performed using Prism (version 10).

## Reporting summary

Further information on research design is available in the Nature Portfolio Reporting Summary linked to this article.

## Data availability

All transgenic *P. knowlesi* lines generated or used in this study are available upon request. These are available from R.W.M. under a material transfer agreement with the Francis Crick Institute and London School of Hygiene and Tropical Medicine. Source data are provided with this paper.

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

## Acknowledgements

M.N.H. was supported by a Bloomsbury Colleges Studentship. F.M. and R.W.M. were supported by the UK Medical Research Council (MRC Career Development Award to R.W.M. MR/M021157/1). J.A.T. was supported by a Wellcome Trust Sir Henry Wellcome Fellowship. G.J.W. and N.M-.S. were funded by the Wellcome Trust (grant 206194). S.M.D. was supported by a UK Medical Research Council LID studentship. H.R.S. was supported by UK Medical Research council Project Grant MR/P010288/1. E.K. was supported by the Francis Crick Institute which receives its core funding from Cancer Research UK (FC001003), the UK Medical Research Council (FC001003), and the Wellcome Trust (FC001003). The authors would like to thank Anthony A. Holder for critical reading of this manuscript and Puck Van Der Laan for illustration of erythrocyte invasion schematics. We would also like to acknowledge the LSHTM Imaging and Cytometry Platform for Infection Biology, with specific thanks to Elizabeth McCarthy for microscopy support.

## Author contributions

M.N.H., R.W.M., and H.R.S. conceived and designed research. M.N.H., F.M., and S.M.D. performed research. M.N.H., R.W.M., J.A.T., F.M. and E.K. contributed to conceptualisation and methodology. N.M-.S. and G.J.W. contributed the generation of recombinant NBPXa for antibody production. M.N.H. and R.W.M. wrote the paper, and all authors contributed to the manuscript and analysed the data.

## Competing interests

The authors declare no competing interests.
