## [Peer Review File · Nature Communications]

nature portfolio

Peer Review FileReviewer comments, first round

Reviewer #1 (Remarks to the Author):

In this manuscript the authors explore the roles of two microneme proteins, PkNBPXa and PkDBP, in the process of RBC invasion by *P. knowlesi* merozoites. *P. knowlesi* provides a useful model to study RBC invasion by merozoites since it is possible to isolate viable, invasive merozoites and perform molecular genetic manipulations to delete or tag genes and produce transgenic parasites for analysis. The authors have used the model to study the localization and movement of PkNBPXa and PkDBP during RBC invasion and define their functional roles in the process of invasion. The study is elegant and provides valuable new information on the roles of these key parasite proteins in the process of host cell invasion. The authors should address the following questions:

1. The experiments studying the invasion of Fluo-4AM labeled RBCs by *P. knowlesi* merozoites show an interesting dual fluorescence signal at the apical end of the merozoites. No major increase in Ca²⁺ is observed in the target RBCs during invasion as observed in case of *P. falciparum*. Is a rise in Ca²⁺ in target RBCs not required for successful invasion by *P. knowlesi*? This is an important difference, which the authors should directly address. For example does *P. knowlesi* invade RBCs pre-loaded with BAPTA-AM? If rise in Ca²⁺ in target RBCs is not required for successful invasion, *P. knowlesi* should be able to invade BAPTA-AM loaded RBCs. This should be tested.
2. The authors interpret the dual fluorescence signal at the apical end of merozoites indicates that the rhoptries fuse with the target RBC membrane, which allows Fluo-4AM to enter the rhoptries. The authors also suggest that the fluorescence signal is due to high levels of Ca²⁺ in the rhoptries. How is Ca²⁺ loaded in the rhoptries and is there any other independent evidence for Ca²⁺ levels being high in the rhoptries? Are Ca²⁺ levels constitutively high in the rhoptries and what might be the role of such high levels for rhoptry function?
3. Video S1: It is not clear on what basis the authors state that pinching and wrapping of the RBC membrane by the invading merozoite goes on from 2 sec to 6.5 sec. It is not clear how the authors draw the temporal boundary for this event. It appears that pinching and wrapping goes on up to 9 secs in the video. Can the authors justify the boundary of 6.5 secs for end of the pinching and wrapping step?
4. Using fluorescently tagged PkNBPXa the authors suggest that PkNBPXa is secreted and spreads over the merozoite surface during invasion. However, fluorescence signal from tagged PkBPXa is not conclusive that the protein is on the surface. To confirm relocalization of PkNBPXa to the merozoite surface, the authors should detect PkNBPXa on the surface of formaldehyde fixed free merozoites using anti-PkNBPXa sera by immunofluorescence assay (IFA) under conditions that don't permeabilize the merozoites.

Reviewer #2 (Remarks to the Author):

Manuscript titled " Sequential roles for red blood cell binding proteins enable phased commitment to invasion for malaria parasites" written by Hart MN et al illustrates two important steps in *Plasmodium knowlesi* merozoites invasion of human and macaque RBCs. Manuscript using genetic knock-out approaches shows that DBP α and RBL, normocyte binding protein Xa are essential for RBC deformities and invasion. I recommend this article for publication as it has described in a systematic fashion two important steps involved in *P. knowlesi* invasion of macaque RBC. I am sure this study will have implications in understanding *P. vivax* invasion of human RBCs.

Manuscript is well written and experiments are sound. A minor suggestion

1. If Authors would have used certain markers such as a anti-MSP-1 abs, anti-GAP 50, anti-AMA-1 Abs as well as abs to other other microneme proteins while delineating the RBC invasion, it would

have given better glimpses of the invasion process

Reviewer #3 (Remarks to the Author):

Review of "Sequential roles for red blood cell binding proteins enable phased commitment to invasion for malaria parasites" by Hart et al for Nature Communications.

Major Comments

Here the roles of two invasion proteins NBPXa and DBPa, members of the RBL/RH and DBP/EBA respectively, are investigated in the parasite *Plasmodium knowlesi*, that can grow in human RBCs. While similar studies have been performed in previously in *P. falciparum*, *P. knowlesi* offers the advantage that the merozoites are larger and more easily visualised by live cell microscopy and have fewer redundant proteins in the RBL and DBP families so firmer conclusions can be drawn with respect to the functions of these proteins. Conditional knockout of NBPXa indicated that the merozoite could still glide but could no longer deform their RBCs. Antibodies blocking DBPa function still permitted strong deformation but the merozoites contacted fewer RBCs. This indicated that NBPXa and DBPa had distinct functions unlike *P. falciparum* where the role of the proteins appears to overlap more. The tagging of both proteins with neogreen indicated they are apically localised in micronemes with NBPXa diffusing out onto the merozoite surface and DBPa having other concentrations at the widest and basal regions of the merozoites. Finally, antibodies to both proteins were found to function synergistically which could inform development of future *knowlesi* and *vivax* vaccines.

I have no major criticisms of this paper and very much enjoyed reading it and found the videos fascinating. I believe this work makes a highly valuable contribution to the study of malaria parasite invasion which is relevant for future vaccine development.

Minor Comments

1. In video S2, the arrows change colour. What does this mean? Different RBCs being contacted?
2. Could a magnified panel be added to Fig 5B.
3. It is mentioned that over time, that the neogreen tagged HBPXa and DBPa proteins become evenly distributed over the mero surface. Could examples be added to Fig 5C.
4. In my version of the paper Figs 5D and E lack enough resolution to be clearly studied.
5. Fig 5F and G are a bit because the text is in minutes and graph is in seconds. Also, it's again too small.

Response to reviewers comments

On behalf of all authors of this paper we would like to thank both the editor and reviewers for their complementary and constructive comments on our manuscript. We were particularly pleased to note that all authors found it recommend the article as both “fascinating” and “elegant”, and broadly interesting to the audience of Nature Communications. Whilst the majority of the issues raised were minor we felt several interesting points warranted some additional experimental work. Specifically, we have added work to demonstrate that BAPTA-AM treatment of host RBCs has no effect on invasion in *P. knowlesi*. We also attempted to use our polyclonal anti-NBPXa antibodies for differential surface/internal labelling immunofluorescence, but were unable to find compatible conditions. Although, the latter was unsuccessful we agree it was a good suggestion and we have amended our manuscript to note this. We have fully addressed the comments point by point below and feel the manuscript has been much improved as a result.

Edits directly in response to reviewers' comments are highlighted in the manuscript in yellow. Edits to correct typos/minor errors spotted when accumulating source data files are highlighted in purple. Additionally, we have modified Figure 7 (previously figure 6) to include updated versions of graphs depicting GIA assay results, due to realising we accidentally plotted all technical repeats, instead of two biological repeats (each with two technical replicates).

REVIEWER COMMENTS

Reviewer #1 (Remarks to the Author):

In this manuscript the authors explore the roles of two microneme proteins, PkNBPXa and PkDBP, in the process of RBC invasion by *P. knowlesi* merozoites. *P. knowlesi* provides a useful model to study RBC invasion by merozoites since it is possible to isolate viable, invasive merozoites and perform molecular genetic manipulations to delete or tag genes and produce transgenic parasites for analysis. The authors have used the model to study the localization and movement of PkNBPXa and PkDBP during RBC invasion and define their functional roles in the process of invasion. The study is elegant and provides valuable new information on the roles of these key parasite proteins in the process of host cell invasion. The authors should address the following questions:

1. The experiments studying the invasion of Fluo-4AM labeled RBCs by *P. knowlesi* merozoites show an interesting dual fluorescence signal at the apical end of the merozoites. No major increase in Ca²⁺ is observed in the target RBCs during invasion as observed in case of *P. falciparum*. Is a rise in Ca²⁺ in target RBCs not required for successful invasion by *P. knowlesi*? This is an important difference, which the authors should directly address. For example does *P. knowlesi* invade RBCs pre-loaded with BAPTA-AM? If rise in Ca²⁺ in target RBCs is not required for successful invasion, *P. knowlesi* should be able to invade BAPTA-AM loaded RBCs. This should be tested. **This is an excellent question, but one we felt initially was outside the scope of this study— simply because we were aiming to use Fluo-4-AM signal as a marker for host cell commitment, rather than exploring the role of calcium in erythrocyte invasion. Weiss et al., (2015) first explored the effects of pre-treating human RBCs with BAPTA-AM. However, Weiss and colleagues observed that *P. falciparum* merozoites could still invade BAPTA treated RBCs (From Weiss et al: “Video microscopy of untreated parasites (W2m) and parasites whose erythrocyte hosts had been treated with BAPTA-AM indicate that chelation of Ca²⁺ introduced during invasion did not reduce the invasion rate in terms of invasion per schizont rupture.”).**

We have repeated this experiment with *P. knowlesi* (according to conditions described in Weiss et al., 2015) and also observe that there is no difference in invasion efficiency

between merozoites invading BAPTA-treated vs. untreated RBCs. It's important to note that since we do not observe any invasions without a Fluo-4 signal (for all events where the merozoite-RBC interface is clearly in focus), this particular "marker" of invasion is very likely essential. However, our BAPTA-AM data indicates that it may not be the case that calcium secretion from the merozoite into the host cell (or leakage into the host cell from surrounding medium) is itself making this step essential. Alternatively, it is also possible that BAPTA-AM fails to chelate incoming calcium quickly enough to result in a noticeable phenotype. With this in mind, we have amended lines 179-188 (copied below with changes highlighted) to describe our new findings:

"Notably, we did not observe any detriment to invasion when human RBCs were pre-loaded with the calcium chelator, BAPTA-AM, in keeping with data for *P. falciparum* invasions (Weiss)(Supplementary Fig X). This result may suggest that a calcium influx into the host cell, whether parasite derived or coming from the surrounding medium, may not be required for invasion, as others have hypothesized (cite them). However, another explanation may be that BAPTA-AM may not chelate incoming calcium quick enough to cause a noticeable phenotype. Importantly, we observed no merozoite detachment after detection of Fluo-4-AM signal or reorientation. This indicates that commitment to invasion, and potentially also to moving junction formation, occurs before reorientation and thus earlier than suggested by data from *Pf* (Fig1E)^{4,26} and furthermore, that Fluo-4 signal can be used as a reliable marker for such commitment."

2. The authors interpret the dual fluorescence signal at the apical end of merozoites indicates that the rhoptries fuse with the target RBC membrane, which allows Fluo-4AM to enter the rhoptries. The authors also suggest that the fluorescence signal is due to high levels of Ca²⁺ in the rhoptries. How is Ca²⁺ loaded in the rhoptries and is there any other independent evidence for Ca²⁺ levels being high in the rhoptries? Are Ca²⁺ levels constitutively high in the rhoptries and what might be the role of such high levels for rhoptry function?

Previous work in Weiss et al. it was proposed this signal arose from "...an open junction between merozoite rhoptries containing Ca²⁺ and erythrocyte cytoplasm containing Fluo-4 could mix at the merozoite apex, indicating a permeabilization or opening of the erythrocyte membrane". Our (clumsy) effort to summarise this with brevity implied that the rhoptries were a well-established Calcium store, when in fact little is known regarding the full repertoire of stores beyond the ER in apicomplexans. We see clear transferral of the dye to an enclosed Ca rich environment in the parasite apex, so we have altered this section to a statement (in lines 176-177) of facts rather than the more speculative statements (removing the sentence "such as the calcium rich rhoptry lumen").

3. Video S1: It is not clear on what basis the authors state that pinching and wrapping of the RBC membrane by the invading merozoite goes on from 2 sec to 6.5 sec. It is not clear how the authors draw the temporal boundary for this event. It appears that pinching and wrapping goes on up to 9 secs in the video. Can the authors justify the boundary of 6.5 secs for end of the pinching and wrapping step? This was an error on our part. We originally had two videos dedicated to showing deformation but must have removed the wrong description of timings from the text. The description should read "(Video S1 from 2.1 sec to 9.0 sec.)". We have altered this in line 126.

4. Using fluorescently tagged PkNBPXa the authors suggest that PkNBPXa is secreted and spreads over the merozoite surface during invasion. However, fluorescence signal from tagged PkNBPXa is not conclusive that the protein is on the surface. To confirm relocation of PkNBPXa to the merozoite surface, the authors should detect PkNBPXa on the surface of formaldehyde fixed free merozoites using anti-PkNBPXa sera by

immunofluorescence assay (IFA) under conditions that don't permeabilize the merozoites. This is a very good suggestion and we felt this would complement our analysis. However, despite repeated attempts with differing conditions we were unable to find a differential labelling protocol compatible with our rabbit anti-NBPXa antibody. We were unable to detect NBPXa under conditions required to show permeabilization/lack of (i.e. no NBPXa signal either before or AFTER permeabilization) due to incompatibilities with fixation approaches. This included when parasites were fixed in solution with 4% PFA (with or without 0.0075% glutaraldehyde) and with or without triton permeabilization. We mention this issue in lines 502-506: "Attempts to localise the NBPXa ectodomain by IFA with a rabbit polyclonal antibody raised against the putative NBPXa RBC-binding domain (amino acids 151-467; Fig6B, schematic) were unsuccessful, as our antibody was not compatible with IFA conditions that permit differentiation between surface vs intracellular localisation." However, we do show by western blot that the NBPXa N-terminus (detectable with our rabbit anti-NBPXa antibody) is shed into culture supernatants, whilst the C-terminal cytoplasmic tail (tagged with a HA-tag) is not. We feel that this strongly supports a model of secretion and subsequent cleavage in line with what is observed for other RBL ligands for other species (eg. Rh5 ligands In Fuvazza et al., 2020 [10.1016/j.chom.2020.02.005](https://doi.org/10.1016/j.chom.2020.02.005)). We have also added a caveat line to the discussion section in (lines 587-590), explaining that dual markers (N-term and C-term tags on a given ligand) would be beneficial to future studies focusing on secretion/cleavage dynamics of RBL ligands: "Future work using double-tagged lines expressing, for instance, a fluorescent marker at the N-terminus of NBPXa in addition to its C-terminus, may yield real-time results that explain the secretion and processing dynamics of RBL ligands more definitively."

Reviewer #2 (Remarks to the Author):

Manuscript titled " Sequential roles for red blood cell binding proteins enable phased commitment to invasion for malaria parasites" written by Hart MN et al illustrates two important steps in Plasmodium knowlesi merozoites invasion of human and macaque RBCs. Manuscript using genetic knock-out approaches shows that DBP α and RBL, normocyte binding protein Xa are essential for RBC deformities and invasion. I recommend this article for publication as it has described in a systematic fashion two important steps involved in P. knowlesi invasion of macaque RBC. I am sure this study will have implications in understanding P. vivax invasion of human RBCs. Manuscript is well written and experiments are sound. A minor suggestion

1. If Authors would have used certain markers such as a anti-MSP-1 abs, anti-GAP 50, anti-AMA-1 Abs as well as abs to other other microneme proteins while delineating the RBC invasion, it would have given better glimpses of the invasion process. **We very much agree. However, very few P. knowlesi specific antibodies are available, at present. Those that cross-react perform poorly when fixed in solution with the paraformaldehyde/glutaraldehyde mixes required to conserve morphological features. Furthermore, in order to study ligand secretion, processing, and localisation during invasion itself, we felt that fluorescent tags would serve us best as we sought to understand these dynamic processes using live microscopy. The transgenic lines generated here are all selection marker free, and so in future work we aim to expand our repertoire of tagged parasite lines to include a wider range of invasion markers and to use faster 3D imaging techniques to build upon the knowledge we present in this study.**

Reviewer #3 (Remarks to the Author):

Review of "Sequential roles for red blood cell binding proteins enable phased commitment to invasion for malaria parasites" by Hart et al for Nature Communications.

Major Comments

Here the roles of two invasion proteins NBPX α and DBP α , members of the RBL/RH and DBP/EBA respectively, are investigated in the parasite *Plasmodium knowlesi*, that can grow in human RBCs. While similar studies have been performed in previously in *P. falciparum*, *P. knowlesi* offers the advantage that the merozoites are larger and more easily visualised by live cell microscopy and have fewer redundant proteins in the RBL and DBP families so firmer conclusions can be drawn with respect to the functions of these proteins. Conditional knockout of NBPX α indicated that the merozoite could still glide but could no longer deform their RBCs. Antibodies blocking DBP α function still permitted strong deformation but the merozoites contacted fewer RBCs. This indicated that NBPX α and DBP α had distinct functions unlike *P. falciparum* where the role of the proteins appears to overlap more. The tagging of both proteins with neogreen indicated they are apically localised in micronemes with NBPX α diffusing out onto the merozoite surface and DBP α having other concentrations at the widest and basal regions of the merozoites. Finally, antibodies to both proteins were found to function synergistically which could inform development of future *knowlesi* and *vivax* vaccines. I have no major criticisms of this paper and very much enjoyed reading it and found the videos fascinating. I believe this work makes a highly valuable contribution to the study of malaria parasite invasion which is relevant for future vaccine development.

Minor Comments

1. In video S2, the arrows change colour. What does this mean? Different RBCs being contacted? **This was indeed quite confusing - apologies! The white arrow points to the merozoite in question, showing the onset of gliding. The red arrow indicates a gliding interaction characterised by deformation, and the blue arrow indicates a gliding interaction without any deformation. We will update the appropriate video legend to include these details.**
2. Could a magnified panel be added to Fig 5B. **We have added a magnified panel, and we have also split Figure 5 into two separate figures (see responses to queries below), as we realise that this was a very full figure, making it difficult to visualise individual panels.**
3. It is mentioned that over time, that the neogreen tagged HBPX α and DBP α proteins become evenly distributed over the mero surface. Could examples be added to Fig 5C. **Further examples of DBP α and NBPX α distribution over the merozoite surface were originally shown in figure 5D and 5E, but these images were too small to be properly studied (apologies!). These images have been significantly enlarged and are now featured in Figure 6A and 6B.**
4. In my version of the paper Figs 5D and E lack enough resolution to be clearly studied. **This was likely in part due to the file processing at submission but these figures were also too small and have now been amended (see answer to above comment)**
5. Fig 5F and G are a bit because the text is in minutes and graph is in seconds. Also, it's again too small. **Graphs are now enlarged, and time references within the text are changed to seconds rather than minutes to match the graphs. "For all conditions tested, peak apical vs peak body intensity normalised (FR approached 1) within ~240 sec after egress" (line 465).**

Reviewer comments, further round

Reviewer #1 (Remarks to the Author):

The authors have addressed my comments/questions.

Reviewer #3 (Remarks to the Author):

The authors have satisfactorily addressed the reviewers' comments and questions.